# Consistency Training with Learnable Data Augmentation for Graph Anomaly Detection with Limited Supervision

**Nan Chen**[1], **Zemin Liu**[1]*, **Bryan Hooi**[1], **Bingsheng He**[1]*, **Rizal Fathony**[2], **Jun Hu**[1], **Jia Chen**[2]

[1]National University of Singapore, [2]GrabTaxi Holdings Pte. Ltd.

```
{chennan, bhooi, hebs}@comp.nus.edu.sg,
{zeminliu, jun.hu}@nus.edu.sg,
{rizal.fathony, jia.chen}@grab.com
```

## Abstract

Graph Anomaly Detection (GAD) has surfaced as a significant field of research, predominantly due to its substantial influence in production environments. Although existing approaches for node anomaly detection have shown effectiveness, they have yet to fully address two major challenges: operating in settings with limited supervision and managing class imbalance effectively. In response to these challenges, we propose a novel model, ConsisGAD, which is tailored for GAD in scenarios characterized by limited supervision and is anchored in the principles of consistency training. Under limited supervision, ConsisGAD effectively leverages the abundance of unlabeled data for consistency training by incorporating a novel learnable data augmentation mechanism, thereby introducing controlled noise into the dataset. Moreover, ConsisGAD takes advantage of the variance in homophily distribution between normal and anomalous nodes to craft a simplified GNN backbone, enhancing its capability to distinguish effectively between these two classes. Comprehensive experiments on several benchmark datasets validate the superior performance of ConsisGAD in comparison to state-of-the-art baselines. Our code is available at https://github.com/Xtra-Computing/ConsisGAD.

## 1 Introduction

Graph Anomaly Detection (GAD) aims to identify abnormal instances or outliers, *e.g.*, nodes, that exhibit behaviors deviating from the norm (Ma et al., 2021). Given the pervasive occurrence of anomalies and their potential negative impact on various applications, GAD has emerged as a prominent research area (Liu et al., 2021b; Shi et al., 2022; Tang et al., 2022; Wang et al., 2023). Owing to the prevalence of class imbalance characteristics (Liu et al., 2023) in GAD, these studies can primarily be categorized into two main approaches: spatial-centric and spectral-centric. Spatial-centric approaches are primarily centered around formulating models by closely analyzing the connecting structure of nodes that require classification, such as dynamically selecting neighboring nodes of the target node (Wang et al., 2019a; Cui et al., 2020; Dou et al., 2020; Liu et al., 2020; 2021a;b), thereby effectively mitigating the impact of imbalanced class distributions. Spectral-centric approaches focus on crafting GNN frameworks equipped with proficient spectral filters (Zhu et al., 2020; Tang et al., 2022; Gao et al., 2023a), to bolster their capacity for improved expressiveness, enabling them to distinguish signals of varying frequencies during neighborhood aggregation.

While existing approaches have demonstrated effectiveness, they still fall short in addressing two substantial challenges. Firstly, these methods frequently require extensive supervision during training, which poses a considerable challenge in scenarios with limited supervision available. Although semi-supervised learning (Van Engelen & Hoos, 2020) offers a remedy by employing high-confidence unlabeled nodes as pseudo-labeled instances, it facilitates label propagation predominantly to nodes showcasing prominent features associated with the labels. This focus inherently

---

*Corresponding authors.

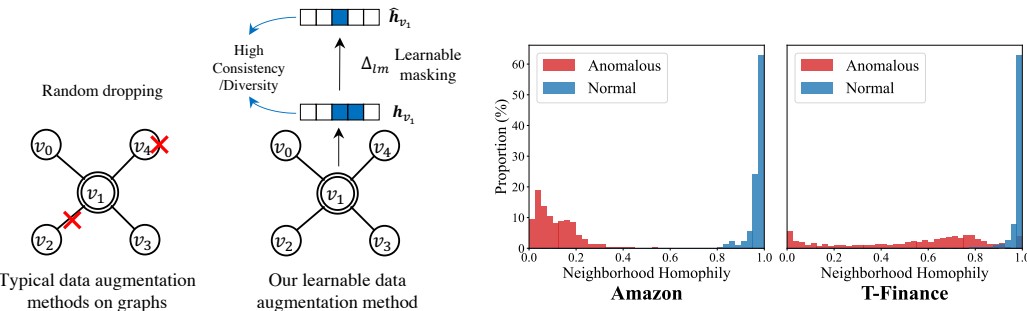

(a) Comparison of the augmentation methods.  (b) The distinction in homophily distribution.

Figure 1: The motivation of our proposed model.

results in the overlooking of a substantial portion of unlabeled instances that bear less distinctive features, rendering label propagation to these instances particularly challenging. While consistency training (Rasmus et al., 2015; Laine & Aila, 2017; Tarvainen & Valpola, 2017) emerges as a promising solution by introducing noise to high-confidence unlabeled nodes—effectively transforming them into less distinctive instances for further consistency-based regularization—its application on graphs typically involves data augmentation through random sampling as the noise (You et al., 2020; Wang et al., 2020; Zhu et al., 2021; Zhao et al., 2021), as shown in Figure 1(a)*(Left)*. This method introduces an inherent difficulty in calibrating the extent of data augmentation and may result in either over-augmentation or under-augmentation (Bo et al., 2022).

Secondly, addressing class imbalance has seen numerous studies typically resorting to reweighting or resampling techniques (Wang et al., 2019a; Cui et al., 2020; Dou et al., 2020; Liu et al., 2020; 2021a;b). They typically aim to dynamically select neighbors of the target node with a view to reducing heterophily and, thereby, alleviating the impact of imbalanced class distributions. However, the inherent uncertainty in predicting neighboring labels considerably influences the reweighting or resampling procedure, often complicating the attainment of preferable homophily neighbors.

To address these challenges, we introduce a novel model, CONSISGAD, designed for Graph Anomaly Detection under limited supervision, grounded in the principles of Consistency Training.

In the context of limited supervision, CONSISGAD harnesses the wealth of unlabeled data for consistency training, through a novel learnable data augmentation mechanism to introduce controlled noise into the dataset, as shown in Figure 1(a)*(Right)*. In particular, we introduce two key metrics, namely, label *consistency* and distribution *diversity* to guide the learning of data augmentation. These metrics assess the relationship between the original data and its augmented version: label consistency emphasizes the retention of identical labels, whereas distribution diversity accentuates the disparities in their representation distributions. This dual-metric approach yields a more appropriate augmentation that shares the same label while exhibiting diverse distribution characteristics, thereby facilitating the label propagating to an extensive space where the features are less distinctive.

To address the class imbalance issue, we argue that the homophily distribution serves as an effective pattern for distinguishing between normal and anomalous nodes. In Figure 1(b), we present statistics on the homophily ratio of each node[1] on two datasets, Amazon and T-Finance (for more details please see Section 4.1). The x-axis represents the edge homophily score, while the y-axis illustrates the proportion of target nodes within the corresponding group (*e.g.*, anomalous or normal). Specifically, normal nodes tend to predominantly associate with other normal neighbors, thereby exhibiting a higher degree of homophily. In contrast, anomalous nodes are often surrounded by a larger proportion of normal neighbors, resulting in lower homophily. This distinction in homophily distribution motivates us to develop a GNN backbone by leveraging the homophily distribution in the context of each target node to effectively discriminate between normal and anomalous nodes.

Additionally, we conduct extensive experiments on four benchmark datasets, alongside one real-world dataset derived from a production environment. The ensuing results highlight the superiority of our proposed CONSISGAD, as it exhibits enhanced performance in comparison to state-of-the-

---

[1]For each target node, we compute the ratio of homophilic edges—those connecting nodes of the same class—to the total count of its neighboring edges.

art approaches. Notably, our GNN model generally outperforms these leading approaches, thereby affirming its efficacy.

## 2 PRELIMINARIES

A graph can be represented as $G = \{V, E, \boldsymbol{X}\}$, where $V$ is the set of nodes, $E$ is the set of edges, and $\boldsymbol{X} \in \mathbb{R}^{|V| \times d_X}$ is the feature matrix of nodes. Let $\boldsymbol{x}_v \in \mathbb{R}^{d_X}$ denotes the feature vector of node $v$. Given a graph encoder $g(\cdot; \theta_g)$ parameterized by $\theta_g$ (*e.g.*, a GNN), we can embed each node $v$ into a low-dimensional representation $\boldsymbol{h}_v \in \mathbb{R}^d$, as $\boldsymbol{h}_v = g(v; \theta_g)$.

**Graph anomaly detection (GAD).** GAD can be conceptualized as a binary classification task, wherein the nodes[2] are classified into one of two categories, namely, normal (the majority) and anomalous (the minority) classes. Formally, given the representation of a node $v$, denoted as $\boldsymbol{h}_v$, a predictor $P(\cdot; \theta_p)$ can be utilized for making predictions, denoted as $\boldsymbol{p}_v \in \mathbb{R}^K$, as follows:

$$\boldsymbol{p}_v = P(\boldsymbol{h}_v; \theta_p) = \text{SOFTMAX}(\boldsymbol{W}_p \boldsymbol{h}_v + \boldsymbol{b}_p), \tag{1}$$

where $\theta_p = \{\boldsymbol{W}_p \in \mathbb{R}^{K \times d}, \boldsymbol{b}_p \in \mathbb{R}^K\}$ are learnable parameters, and $K = 2$ in our case of anomaly detection. For a labeled node $v$, let $\boldsymbol{y}_v \in \mathbb{R}^K$ denotes the one-hot label vector, where $\boldsymbol{y}_v[k] = 1$ if and only if node $v$ belongs to class $k$. Typically, given the training set $V_{tr}$, the loss function is formulated by applying cross-entropy loss to the predictions, as outlined below.

$$\mathcal{L} = -\sum_{v \in V_{tr}} \sum_{k=0}^{K-1} \boldsymbol{y}_v[k] \ln \boldsymbol{p}_v[k]. \tag{2}$$

**Consistency training.** In the graph setting, consistency training (Wang et al., 2020) involves introducing noise to high-quality unlabeled nodes to generate their augmentations. This allows for the application of consistency-based regularization between the original and augmented versions, thereby assisting in the training of the main model through the enhancement of label propagation. Specifically, to identify high-quality nodes, a given unlabeled node $v$ and its prediction $\boldsymbol{p}_v \in \mathbb{R}^K$ can be assessed using a threshold $\tau$, based on their predicted scores. Formally, given the set of unlabeled nodes $V_{un}$, the high-quality nodes can be defined as $V_{hq} = \{v \mid v \in V_{un} \wedge \exists \boldsymbol{p}_v[k] \geq \tau\}$. For each high-quality node $v \in V_{hq}$, we can compute its one-hot predicted pseudo label vector $\tilde{\boldsymbol{y}}_v$, where $\tilde{\boldsymbol{y}}_v[\arg\max \boldsymbol{p}_v] = 1$, and the other elements are zeros. In this scenario, consistency-based regularization is enforced between the original and augmented nodes, as described below.

$$\mathcal{L}_c = -\sum_{v \in V_{hq}} \sum_{k=0}^{K-1} \tilde{\boldsymbol{y}}_v[k] \ln \boldsymbol{p}_{\hat{v}}[k], \tag{3}$$

where $\boldsymbol{p}_{\hat{v}}$ represents the prediction of the augmented version $\hat{v}$ as per Equation (1). By optimizing the model w.r.t. the combined loss, denoted as $\mathcal{L} + \mathcal{L}_c$, the model benefits from supplementary guidance provided by the consistency regularization, enhancing label propagation and overall performance.

## 3 THE PROPOSED MODEL: CONSISGAD

In this section, we present the proposed CONSISGAD for graph anomaly detection, with the overall framework depicted in Figure 2. The CONSISGAD comprises two principal components: consistency training and the training of the learnable data augmentation module. On one hand, given the availability of both labeled and unlabeled data, we formulate consistency training by leveraging the learnable data augmentation module to generate superior augmentations. On the other hand, with the unlabeled nodes at our disposal, we optimize the learnable data augmentation module with respect to the proposed consistency and diversity loss, aiming to adaptively produce ideal augmentations.

In the following part, we first introduce our backbone GNN model (Section 3.1), specifically designed to effectively address the class imbalance issue inherent in anomaly detection. Following this, we delve into the details of consistency training with learnable data augmentation (Section 3.2), a mechanism instrumental in solving anomaly detection in the scenario with limited supervision.

---

[2]In this paper, we focus solely on the node-level graph anomaly detection.

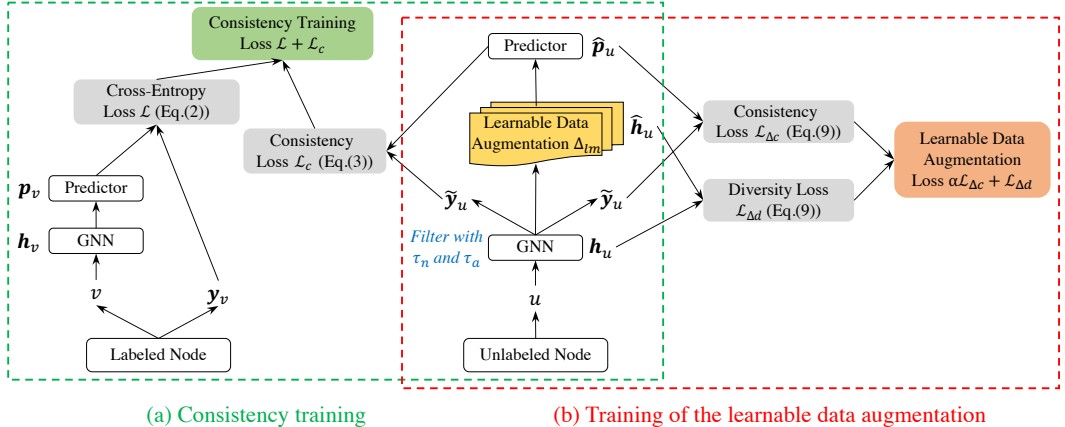

(a) Consistency training        (b) Training of the learnable data augmentation

Figure 2: Overall framework of CONSISGAD.

## 3.1 HOMOPHILY-AWARE NEIGHBORHOOD AGGREGATION

As discussed in the Introduction, the significant difference in contextual homophily distribution between normal and anomalous nodes inspires the formulation of the GNN backbone. This backbone capitalizes on the homophily distribution within the context of a node, thereby enhancing the capability to distinguish between normal and anomalous nodes to deal with the imbalance issue.

Specifically, for a node $v$ and its neighborhood $\mathcal{N}_v$, we first compute the homophily representation between $v$ and each of its neighbors. In other words, we determine the homophily representation along each edge, to establish the groundwork for calculating the contextual homophily distribution, which can be achieved by aggregating the edge-level homophily representation to finally serve as the node representation. Formally, in the $l^{th}$ layer, this node representation $\boldsymbol{h}_v^l$ can be expressed as

$$\boldsymbol{h}_v^l = \text{AGGR}\Big\{\delta(\boldsymbol{h}_v^{l-1}, \boldsymbol{h}_u^{l-1}; \theta_\delta) : u \in \mathcal{N}_v\Big\}. \tag{4}$$

Here $\delta(\cdot, \cdot; \theta_\delta)$ is a function parameterized by $\theta_\delta$ to calculate the edge-level homophily representation, and we instantiate it as $\text{MLP}(\boldsymbol{h}_v^{l-1}||\boldsymbol{h}_u^{l-1})$. $\text{AGGR}(\cdot)$ is the aggregation function, such as sum operator. In the following, we still use $\boldsymbol{h}_v$ to denote the output embedding of node $v$ for simplicity.

**Analysis.** Basically, the edge-level homophily representation can depict the homophily relationship between the two terminal nodes of a given edge. Given the variability of homophily across different edges, it becomes imperative to represent the contextual homophily representation from the perspective of each individual edge. The aggregation of these edge-level homophily representations subsequently illustrates the overall contextual homophily distribution.

To assess its efficacy, we train the entire CONSISGAD and exclusively visualize the intermediate edge-level homophily representations via T-SNE, calculated by the function $\delta(\cdot, \cdot; \theta_\delta)$, on two datasets: Amazon and T-Finance. The visualization results are depicted in Figure 3. Specifically, for the heterogeneous graph Amazon, there exist three types of relations, each of which is visualized in a separate subfigure. It is important to note that four colors are used in each figure to represent the target-neighbor type, namely AN (Anomalous-Normal), NA (Normal-Anomalous), AA (Anomalous-Anomalous), and NN (Normal-Normal). The visualizations reveal that the edge-level homophily representation can adeptly mirror the type of edge homophily, thereby providing a solid foundation for aggregation to represent the contextual homophily distribution of each node. Additionally, our experimental results, detailed in Section 4.2.1, demonstrate that our proposed backbone, utilizing homophily-aware neighborhood aggregation, can attain performance that is comparable to, or even surpasses, that of state-of-the-art approaches in the realm of graph anomaly detection.

## 3.2 CONSISTENCY TRAINING WITH LEARNABLE DATA AUGMENTATION

To improve the performance of graph anomaly detection under limited supervision, we incorporate consistency training (Rasmus et al., 2015; Laine & Aila, 2017), which enables us to harness the

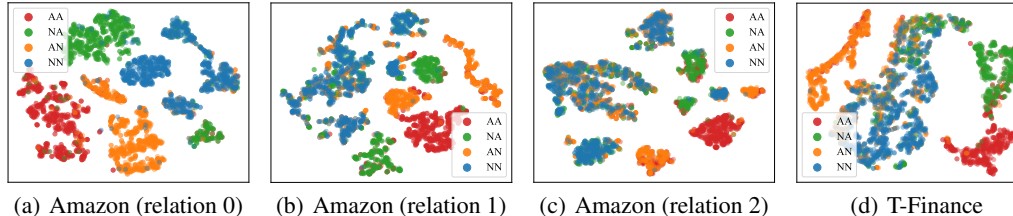

| (a) Amazon (relation 0) | (b) Amazon (relation 1) | (c) Amazon (relation 2) | (d) T-Finance |

Figure 3: Visualization of edge-level homophily distribution: subfigures (a), (b), and (c) depict the three relations on Amazon, while (d) depicts T-Finance.

inherent information embedded within the graph. This is achieved through the use of data augmentation for noise injection (Xie et al., 2020). Given the significant impact of data augmentation on the effectiveness of model training, formulating advanced augmentation strategies is of paramount importance. For example, in classification tasks, an ideal augmentation should preserve essential information that enables the correct categorization of the augmented data, while simultaneously introducing diverse elements not explicitly related to the class. This incorporation of diversity serves to expand the representation space, thereby assisting in simulating less distinctive instances for label propagation (Xie et al., 2020). In the subsequent section, we will first introduce the definitions of the evaluation metrics, and further discuss the concept of learnable data augmentation.

### 3.2.1 LABEL CONSISTENCY AND DISTRIBUTION DIVERSITY

To synthesize the appropriate augmentations, we draw inspiration from (Bo et al., 2022) to design two metrics, namely, label *consistency* and distribution *diversity*, to quantitatively evaluate the augmentations. Given a noise injection method (*e.g.*, data augmentation) $\Delta(\cdot; \theta_\Delta)$, a noised version of the unlabeled node $v \in V_{un}$ can be defined as $\hat{v} = \Delta(v; \epsilon, \theta_\Delta)$, where a small noise $\epsilon$ is injected. Consequently, with the graph encoder $g(\cdot; \theta_g)$, the representation and predicted label vector of the synthetic node $\hat{v}$ can be represented as $\boldsymbol{h}_{\hat{v}} = g(\hat{v}; \theta_g)$ and $\tilde{\boldsymbol{y}}_{\hat{v}}$, respectively. In this case, the label *consistency* between the original version $v$ and its noised version $\hat{v}$ can be formalized as

$$C(v, \hat{v}) = \mathbb{I}(\tilde{\boldsymbol{y}}_v = \tilde{\boldsymbol{y}}_{\hat{v}}), \tag{5}$$

where $\mathbb{I}(\cdot) = 1$ if the inside condition holds, otherwise $\mathbb{I}(\cdot) = 0$. On the other hand, the distribution *diversity* between them can be formalized with their representations, as

$$D(v, \hat{v}) = d(\boldsymbol{h}_v, \boldsymbol{h}_{\hat{v}}), \tag{6}$$

where $d(\cdot, \cdot)$ is a distance function defined on the vector space, *e.g.*, Euclidean distance. Ideally, an effective augmentation approach should synthesize augmented instances that maintain high consistency with the original instances while involving as much diversity as possible.

While (Bo et al., 2022) also propose metrics for consistency and diversity, our definitions differ from theirs in the following aspects: (1) Our metrics directly operate on the given data and its augmentations, and do not require a validation set for their evaluation, a necessary component in (Bo et al., 2022). (2) Our metrics can further guide the learnable data augmentation module to synthesize preferable data augmentations, which will be discussed in Section 3.2.2. In contrast, their approach only aims to select suitable augmentations from pre-defined candidates.

### 3.2.2 LEARNABLE DATA AUGMENTATION

Existing data augmentation techniques for graphs often rely on hand-crafted or random modifications to the original data, such as node dropping (Feng et al., 2020b), edge dropping (Rong et al., 2020), feature masking (You et al., 2020), *etc.* As discussed in the Introduction, these approaches have difficulty in calibrating the extent of data augmentation and can lead to over- or under-augmentations, hindering label propagation within the label space on the graph. To generate ideal augmentations, a promising strategy is to make the process learnable, using the raw instance as well as the evaluation metrics including label consistency and distribution diversity.

To achieve this, we employ perturbations on the intermediate states as a means of introducing noise to the data. This approach is a straightforward method for adding noise to hidden tensors and has been demonstrated to be as effective as other augmentation strategies on graphs (Xia et al., 2022).

Initially, we refine the definition of high-quality nodes $V_{hq}$ (defined in Section 2), in the context of the anomaly detection setting. Due to the class imbalance issue, normal and anomalous nodes tend to have distinct confidence levels in their prediction scores. Therefore, we assign a separate threshold for each class, specifically, $V_{hq} = \{v \mid v \in V_{un} \land \boldsymbol{p}_v[0] \geq \tau_n\} \cup \{v \mid v \in V_{un} \land \boldsymbol{p}_v[1] \geq \tau_a\}$, where $\tau_n$ and $\tau_a$ represent the thresholds for the normal and anomalous classes[3], respectively. In Appendix E.6, we present a detailed evaluation of the quality of selected high-quality nodes.

Subsequently, given the representation of a high-quality node $v \in V_{hq}$—denoted as $\boldsymbol{h}_v$—we introduce a learnable data augmentation module to synthesize the augmented version of its representation, denoted as $\hat{\boldsymbol{h}}_v$, through *learnable masking*, strategically preserving the dimensions exhibiting high consistency and diversity while omitting the others. Formally, $\hat{\boldsymbol{h}}_v$ can be expressed as:

$$\hat{\boldsymbol{h}}_v = \Delta_{lm}(\boldsymbol{h}_v; \theta_{\Delta_{lm}}), \tag{7}$$

where $\Delta_{lm}(\cdot; \theta_{\Delta_{lm}})$, parameterized by $\theta_{\Delta_{lm}}$, represents the learnable masking module, generating the augmented representation for an input node. To effectively select the appropriate dimensions for preservation and masking, given the input vector $\boldsymbol{h}_v$, we utilize the attention mechanism to calculate the weight for each dimension of $\boldsymbol{h}_v$, subsequently sharpening the weights into zeros and ones based on a predefined masking rate. Formally, function $\Delta_{lm}(\cdot; \theta_{\Delta_{lm}})$ can be instantiated as:

$$\hat{\boldsymbol{h}}_v = \Delta_{lm}(\boldsymbol{h}_v; \theta_{\Delta_{lm}}) = \text{SHARPEN}(\text{ATTEN}(\boldsymbol{h}_v); \xi) \odot \boldsymbol{h}_v. \tag{8}$$

Here, $\text{ATTEN}(\boldsymbol{h})$ denotes the attention function, taking the representation to be augmented as input and outputting a vector of the same dimension, indicating the importance of each dimension. A simplified version of this attention module can be represented as $\text{ATTEN}(\boldsymbol{h}) = \boldsymbol{W}\boldsymbol{h} + \boldsymbol{b}$, with $\boldsymbol{W} \in \mathbb{R}^{d \times d}$ and $\boldsymbol{b} \in \mathbb{R}^d$ being the weight matrix and bias vector, respectively. $\text{SHARPEN}(\boldsymbol{h}; \xi)$ is a function designed to retain the dimensions with the highest values in $\boldsymbol{h}$ as ones and the rest as zeros, by constraining the proportion of zeros with ratio $\xi$. For instance, given vector $\boldsymbol{a} = [0.1, 0.4, 0.2, 0.3]$ and $\xi = 0.5$, $\text{SHARPEN}(\boldsymbol{a}; \xi) = [0, 1, 0, 1]$. It is noteworthy that in our implementation this sharpen function does not truncate gradients, ensuring uninterrupted backpropagation and facilitating end-to-end model training. The pseudocode for the sharpen function is provided in the Appendix A.

To facilitate learnable augmentation, the augmented representations are subsequently used to compute the consistency and diversity loss, adhering to the strategies outlined in Equations (5) and (6). Formally, for a high-quality node $v \in V_{hq}$, we initially compute the predictions $\boldsymbol{p}_v$ in accordance with Equation (1), as well as the pseudo label vector $\tilde{\boldsymbol{y}}_v$, where $\tilde{\boldsymbol{y}}_v[\arg\max \boldsymbol{p}_v] = 1$. Conversely, for the augmented representation $\hat{\boldsymbol{h}}_v$, we also calculate the prediction w.r.t. Equation (1) as $\hat{\boldsymbol{p}}_v = P(\hat{\boldsymbol{h}}_v; \theta_p)$. Subsequently, both the original and the augmented predictions are leveraged to formulate the consistency and diversity loss, namely, $\mathcal{L}_{\Delta c}$ and $\mathcal{L}_{\Delta d}$, as follows:

$$\mathcal{L}_{\Delta c} = -\sum_{v \in V_{hq}} \sum_{k=0}^{K-1} \tilde{\boldsymbol{y}}_v[k] \ln \hat{\boldsymbol{p}}_v[k], \qquad \mathcal{L}_{\Delta d} = -\sum_{v \in V_{hq}} d(\boldsymbol{h}_v, \hat{\boldsymbol{h}}_v). \tag{9}$$

Finally, we combine the two losses with a weight factor $\alpha$, *i.e.*, $\alpha\mathcal{L}_{\Delta c} + \mathcal{L}_{\Delta d}$, to guide the training of the learnable data augmentation, as shown in Figure 2(b). Specifically, the training of this module is guided by this fused consistency and diversity loss, rather than the consistency training loss. This is because an ideal augmentation should maintain high consistency and diversity with the original version (shown in Figure 2(b)), which is unrelated to the consistency training loss in Figure 2(a). Only the synthesized augmentations can subsequently serve as input for the consistency training.

It is imperative to note that although $\mathcal{L}_{\Delta c}$ (Equation (9)) and $\mathcal{L}_c$ (Equation (3)) exhibit similar forms, their objectives are different: $\mathcal{L}_{\Delta c}$ aims to optimize the learnable data augmentation module to synthesize the ideal augmentations, whereas $\mathcal{L}_c$ is utilized for consistency training in conjunction with the main objective $\mathcal{L}$ (see Equation (2)). To elucidate this distinction further, we will elaborate on the training paradigm in the subsequent section.

## 3.3 TRAINING PARADIGM

As depicted in Figure 2, CONSISGAD incorporates two main components: (a) consistency training, and (b) training of the learnable data augmentation module. Consistency training is centered on

---

[3]We assume the first and second dimensions correspond to the normal and anomalous classes, respectively.

refining the GNN model to generate enhanced representations, with parameters of GNN being exclusively optimized through this process. In contrast, the training of the learnable data augmentation module is directed towards synthesizing optimal augmentations, instrumental for the consistency training. Therefore, the optimization of this module is solely driven by the consistency and diversity loss. Consequently, these two components—consistency training and the training of the learnable data augmentation module—are trained iteratively, with one component being optimized while the other remains fixed. Specifically, the optimization of the learnable data augmentation contributes to the training of the GNN model by providing optimal augmentations. Simultaneously, a proficiently trained GNN encoder, based on consistency training, lays the groundwork for generating superior augmentations. The training procedure and its complexity analysis are detailed in Appendix A.

## 4 EXPERIMENTS

### 4.1 EXPERIMENTAL SETUP

**Datasets.** We conduct experiments on four benchmark GAD datasets, namely Amazon (McAuley & Leskovec, 2013), YeclpChi (Rayana & Akoglu, 2015), T-Finance (Tang et al., 2022), and T-Social (Tang et al., 2022), as summarized in Table 3. Additionally, we perform experiments on an industrial graph from Grab, a leading superapp in Southeast Asia. Detailed descriptions are in Appendix B.

**Baselines.** We employ state-of-the-art approaches from two main categories for comparison. (1) *Generic GNN models*, including MLP (Rosenblatt, 1958), GCN (Kipf & Welling, 2017), GraphSAGE (Hamilton et al., 2017), GAT (Veličković et al., 2018), GIN (Xu et al., 2019), and GATv2 (Brody et al., 2022); and (2) *GAD approaches*, including CARE-GNN (Dou et al., 2020), GraphConsis (Liu et al., 2020), PC-GNN (Liu et al., 2021b), BWGNN (Tang et al., 2022), H2-FDetector (Shi et al., 2022), GHRN (Gao et al., 2023a), GDN (Gao et al., 2023b), and GAGA (Wang et al., 2023). Additionally, we also assess our proposed GNN backbone as presented in Section 3.1, deploy it in the same training paradigm as that of generic GNN models, and refer it as CONSISGAD(GNN). For detailed descriptions of these baselines, please kindly refer to Appendix C.

**Experimental setup.** In our major experiments, we focus on settings with limited supervision. Depending on the dataset size, we set the training ratio to $1\%$ for Amazon, YelpChi, T-Finance, and Industrial graph, and $0.01\%$ for T-Social. In all scenarios, the remaining data is split in a 1:2 ratio for validation and testing, while all data is utilized as unlabeled for consistency training. We adopt AUROC, AUPRC, and Macro F1, to assess model performance. The average score and standard deviation across five independent runs are reported. More details of the settings are in Appendix D.

### 4.2 PERFORMANCE EVALUATION AND ANALYSIS

#### 4.2.1 GRAPH ANOMALY DETECTION

**With limited supervision.** We present the performance comparison on four benchmark datasets with a 1% training ratio in Table 1 (Amazon and YelpChi) and Table 2 (T-Finance and T-Social), and the comparision on industrial data in Appendix E.1. Our proposed CONSISGAD demonstrates superior performance in most cases, underscoring its efficacy in graph anomaly detection. The only deviation is observed in the Amazon dataset, where BWGNN(homo) registers the highest Macro F1, and CONSISGAD closely follows as the runner-up. This discrepancy may stem from BWGNN's reliance on Macro F1 to select its optimal models, potentially neglecting the other two metrics where CONSISGAD excels. Notably, our proposed GNN backbone, CONSISGAD(GNN), surpasses all baselines on YelpChi, T-Finance, and T-Social datasets, as indicated in blue, and maintains competitive performance on the Amazon dataset. This underscores the ability of our backbone GNN to capture the homophily distribution within the context of each node for anomaly detection. Furthermore, the full model CONSISGAD consistently outperforms CONSISGAD(GNN), highlighting the effectiveness of consistency training coupled with learnable data augmentation.

In addition, we observe that models explicitly designed for GAD generally outperform classic GNN models across the datasets. Specifically, spectral-based models like BWGNN and GHRN tend to exhibit superior performance. In the Amazon dataset, node feature-oriented methods such as MLP and GDN yield promising results. This might be attributed to node features playing a more pivotal role than connections in representing nodes on this specific dataset.

| Methods | Amazon | | | YelpChi | | |
|---|---|---|---|---|---|---|
| | AUROC | AUPRC | Macro F1 | AUROC | AUPRC | Macro F1 |
| MLP | $92.39_{\pm0.72}$ | $79.37_{\pm1.83}$ | $87.53_{\pm1.61}$ | $72.18_{\pm0.39}$ | $31.09_{\pm0.52}$ | $61.61_{\pm0.33}$ |
| GCN | $87.34_{\pm0.59}$ | $48.06_{\pm2.73}$ | $70.94_{\pm2.43}$ | $54.65_{\pm0.53}$ | $17.07_{\pm0.44}$ | $35.59_{\pm10.27}$ |
| GraphSAGE | $90.12_{\pm0.48}$ | $73.17_{\pm4.65}$ | $84.25_{\pm2.26}$ | $73.70_{\pm0.52}$ | $34.57_{\pm0.78}$ | $63.33_{\pm0.51}$ |
| GAT | $80.74_{\pm3.64}$ | $45.46_{\pm11.09}$ | $63.45_{\pm12.82}$ | $70.14_{\pm1.91}$ | $28.90_{\pm1.98}$ | $61.22_{\pm1.32}$ |
| GIN | $84.35_{\pm0.75}$ | $39.96_{\pm2.00}$ | $71.20_{\pm1.37}$ | $56.98_{\pm0.82}$ | $18.34_{\pm0.64}$ | $53.58_{\pm0.41}$ |
| GATv2 | $85.39_{\pm3.19}$ | $62.74_{\pm15.12}$ | $76.20_{\pm13.69}$ | $72.83_{\pm0.75}$ | $31.87_{\pm1.47}$ | $62.23_{\pm0.56}$ |
| CARE-GNN | $89.68_{\pm0.76}$ | $50.56_{\pm3.96}$ | $75.74_{\pm0.50}$ | $72.11_{\pm1.23}$ | $31.09_{\pm1.71}$ | $61.62_{\pm0.87}$ |
| GraphConsis | $64.23_{\pm13.83}$ | $21.38_{\pm10.37}$ | $55.35_{\pm8.09}$ | $\underline{78.91}_{\pm0.88}$ | $38.40_{\pm2.63}$ | $64.96_{\pm2.54}$ |
| PC-GNN | $91.18_{\pm0.66}$ | $77.92_{\pm1.49}$ | $85.25_{\pm2.09}$ | $75.17_{\pm0.44}$ | $36.60_{\pm0.91}$ | $64.23_{\pm0.47}$ |
| BWGNN(homo) | $88.56_{\pm0.87}$ | $79.26_{\pm1.11}$ | $\mathbf{90.48}_{\pm0.98}$ | $72.15_{\pm0.59}$ | $30.93_{\pm1.48}$ | $61.24_{\pm0.39}$ |
| BWGNN(hetero) | $84.64_{\pm1.31}$ | $64.00_{\pm7.00}$ | $80.41_{\pm4.29}$ | $77.62_{\pm2.37}$ | $\underline{39.87}_{\pm1.79}$ | $\underline{66.54}_{\pm0.73}$ |
| H2-FDetector | $83.81_{\pm2.57}$ | $49.37_{\pm4.46}$ | $72.46_{\pm4.01}$ | $72.38_{\pm1.13}$ | $32.37_{\pm3.25}$ | $63.97_{\pm0.66}$ |
| GHRN(homo) | $88.35_{\pm2.03}$ | $74.50_{\pm4.57}$ | $86.35_{\pm2.60}$ | $72.03_{\pm0.90}$ | $30.76_{\pm1.24}$ | $61.32_{\pm0.37}$ |
| GHRN(hetero) | $84.40_{\pm2.77}$ | $60.84_{\pm9.80}$ | $78.81_{\pm4.45}$ | $75.33_{\pm1.44}$ | $34.53_{\pm2.83}$ | $63.62_{\pm1.41}$ |
| GDN | $92.16_{\pm0.12}$ | $\underline{81.87}_{\pm0.17}$ | $89.75_{\pm0.05}$ | $75.92_{\pm0.51}$ | $38.04_{\pm0.83}$ | $64.81_{\pm0.25}$ |
| GAGA | $82.61_{\pm6.87}$ | $56.59_{\pm6.60}$ | $76.85_{\pm8.08}$ | $71.61_{\pm2.13}$ | $31.96_{\pm3.37}$ | $61.81_{\pm1.69}$ |
| CONSISGAD(GNN) | $92.01_{\pm0.71}$ | $78.49_{\pm0.40}$ | $85.53_{\pm0.51}$ | $80.95_{\pm0.36}$ | $43.25_{\pm0.31}$ | $67.62_{\pm0.31}$ |
| CONSISGAD | $\mathbf{93.91}_{\pm0.58}$ | $\mathbf{83.33}_{\pm0.34}$ | $\underline{90.03}_{\pm0.53}$ | $\mathbf{83.36}_{\pm0.53}$ | $\mathbf{47.33}_{\pm0.58}$ | $\mathbf{69.72}_{\pm0.30}$ |

Table 1: Comparison (%) on Amazon and YelpChi, with the best bolded and runner-up underlined.

| Methods | T-Finance | | | T-Social | | |
|---|---|---|---|---|---|---|
| | AUROC | AUPRC | Macro F1 | AUROC | AUPRC | Macro F1 |
| MLP | $92.17_{\pm0.64}$ | $52.79_{\pm5.41}$ | $82.33_{\pm0.54}$ | $66.95_{\pm0.71}$ | $6.00_{\pm0.33}$ | $54.09_{\pm0.61}$ |
| GCN | $89.29_{\pm0.19}$ | $53.94_{\pm3.22}$ | $77.16_{\pm1.20}$ | $83.30_{\pm1.60}$ | $23.79_{\pm2.43}$ | $65.16_{\pm0.92}$ |
| GraphSAGE | $89.42_{\pm1.36}$ | $49.08_{\pm6.34}$ | $77.62_{\pm1.87}$ | $71.45_{\pm2.24}$ | $8.73_{\pm0.91}$ | $56.47_{\pm0.64}$ |
| GAT | $87.40_{\pm4.41}$ | $45.60_{\pm14.62}$ | $75.49_{\pm5.63}$ | $73.46_{\pm3.32}$ | $13.47_{\pm2.83}$ | $61.98_{\pm2.06}$ |
| GIN | $81.29_{\pm1.66}$ | $21.66_{\pm3.98}$ | $65.38_{\pm3.05}$ | $78.70_{\pm2.19}$ | $16.24_{\pm5.53}$ | $61.62_{\pm5.93}$ |
| GATv2 | $73.25_{\pm10.00}$ | $18.70_{\pm15.26}$ | $63.16_{\pm9.02}$ | $79.89_{\pm4.80}$ | $16.74_{\pm2.10}$ | $62.99_{\pm1.34}$ |
| CARE-GNN | $91.45_{\pm0.40}$ | $72.27_{\pm1.09}$ | $83.68_{\pm0.78}$ | - OOM - | - OOM - | - OOM - |
| GraphConsis | $92.61_{\pm0.47}$ | $70.70_{\pm1.85}$ | $85.37_{\pm0.48}$ | - OOM - | - OOM - | - OOM - |
| PC-GNN | $91.74_{\pm0.85}$ | $74.77_{\pm0.98}$ | $\underline{86.97}_{\pm0.24}$ | $64.68_{\pm0.64}$ | $4.30_{\pm0.09}$ | $49.66_{\pm0.12}$ |
| BWGNN | $\underline{93.08}_{\pm1.57}$ | $\underline{77.79}_{\pm3.87}$ | $86.97_{\pm1.51}$ | $84.40_{\pm3.01}$ | $49.96_{\pm3.75}$ | $76.37_{\pm1.82}$ |
| H2-FDetector | - OOM - | - OOM - | - OOM - | - OOM - | - OOM - | - OOM - |
| GHRN | $91.93_{\pm0.93}$ | $65.94_{\pm4.38}$ | $80.05_{\pm4.43}$ | $84.20_{\pm3.91}$ | $37.04_{\pm11.02}$ | $71.25_{\pm4.32}$ |
| GDN | $88.75_{\pm1.79}$ | $54.27_{\pm4.31}$ | $76.62_{\pm3.90}$ | $67.69_{\pm1.49}$ | $7.51_{\pm0.79}$ | $55.76_{\pm0.82}$ |
| GAGA | $92.36_{\pm1.45}$ | $64.34_{\pm6.01}$ | $81.10_{\pm2.60}$ | $78.92_{\pm1.26}$ | $23.72_{\pm4.81}$ | $65.58_{\pm3.30}$ |
| CONSISGAD(GNN) | $94.72_{\pm0.11}$ | $83.92_{\pm0.15}$ | $89.73_{\pm0.38}$ | $93.54_{\pm0.35}$ | $53.40_{\pm1.28}$ | $76.45_{\pm1.06}$ |
| CONSISGAD | $\mathbf{95.33}_{\pm0.30}$ | $\mathbf{86.63}_{\pm0.44}$ | $\mathbf{90.97}_{\pm0.63}$ | $\mathbf{94.31}_{\pm0.20}$ | $\mathbf{58.38}_{\pm2.10}$ | $\mathbf{78.08}_{\pm0.54}$ |

Table 2: Comparison (%) on T-Finance and T-Social, with the best bolded and runner-up underlined. Here, "- OOM -" means the Out Of GPU Memory issue when running the model.

**Under varying supervision.** We extend our comparison to include varying supervision levels, specifically 1%, 3%, 5%, 10%, and 40%, focusing on two datasets and the most competitive baselines: BWGNN, GHRN, GDN, and GAGA. For Amazon and YelpChi, we employ the homo and heter versions of BWGNN and GHRN, respectively, owing to their superior performance in previous experiments. The AUPRC metric results are depicted in Figure 4. We observe that CONSISGAD surpasses its competitors at training ratios up to 10%, showcasing its efficacy in graph anomaly detection under limited supervision. This is attributed to the robust GNN backbone and the consistency training with learnable data augmentation. A comparison between CONSISGAD and our backbone reveals that the performance gap narrows with an increase in training data. This aligns with expectations, as the availability of more labeled data tends to diminish the impact of consistency training and learnable data augmentation. For results related to the other two metrics, kindly refer to Appendix E.2, which also shows similar trends.

### 4.2.2 MODEL ANALYSIS

**Ablation study: influence of label consistency and distribution diversity.** To examine the contributions of label consistency and distribution diversity to the training of CONSISGAD, we con-

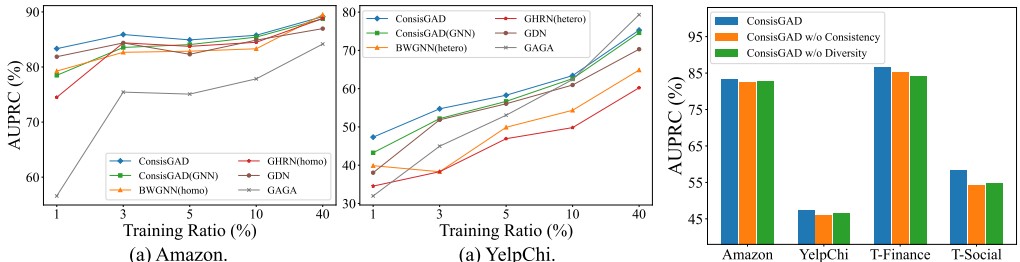

Figure 4: Experiments under varying supervision.  Figure 5: Ablation study.

ducted an ablation study, omitting each element while keeping the rest of the model intact. The results, focusing on the AUPRC metric, are depicted in Figure 5, with similar patterns observed for the other metrics available in Appendix E.3. A performance decline is noted upon the removal of either component, underscoring their synergistic impact on augmentation quality and, consequently, the consistency training process. Predominantly, label consistency emerges as more crucial, adhering to the fundamental principle of preserving label consistency in data augmentation.

**Additional analysis.** Due to space constraints, additional analyses are relegated to the Appendix. Included are comparisons with traditional stochastic graph augmentation methods (Appendix E.4), an exploration of our method's adaptability across various GNN architectures (Appendix E.5), an examination of our model's sensitivity to hyper-parameters (Appendix E.7), and a study on the impact of exact labels on consistency training (Appendix E.9). Furthermore, we analyze the performance of CONSISGAD(GNN) in generic multi-class node classification tasks in Appendix E.8.

## 5 RELATED WORK

**Graph Anomaly Detection.** Research on GAD can be broadly categorized into spatial-centric and spectral-centric methods. Spatial-centric techniques primarily address the class imbalance issue by segregating the neighbors into homophilous and heterophilous sub-groups and employing distinct message-passing strategies for each (Dou et al., 2020; Liu et al., 2020; 2021b; Dong et al., 2022; Huang et al., 2022; Shi et al., 2022; Wang et al., 2023; Zhuo et al., 2024), or other angles (Zhang et al., 2021; Li et al., 2022; Qin et al., 2022; Gao et al., 2023b). On the other hand, spectral-centric methods arise from the observation that the class imbalance leads to high-frequency signals in the graph spectral domain, often utilizing band-pass spectral filters to identify these anomalous signals (Tang et al., 2022; Gao et al., 2023a). However, these methods often overlook scenarios with limited supervision, assuming abundant supervision is available. Furthermore, a majority of existing works opt to mitigate the heterophily issue rather than harness neighborhood homophily to enhance predictions.

**Consistency training.** Consistency training (Bachman et al., 2014) enforces the stability of model outputs with regard to perturbed inputs. Various extensions have been proposed to exploit unlabeled data in fields like vision (Miyato et al., 2018; Berthelot et al., 2019; 2020; Xie et al., 2020; Sohn et al., 2020; Berthelot et al., 2022), language (Liang et al., 2018; Clark et al., 2018), or graph (Deng et al., 2019; Feng et al., 2019; Wang et al., 2020; Feng et al., 2020a; Verma et al., 2021). However, we are the first one to utilize consistency training for graph anomaly detection with limited supervision.

**Data augmentation on graphs.** We delegate to Appendix F the discussion of our method within the literature of data augmentation on graphs, with a further comparison between existing automatic data augmentation techniques with our own.

## 6 CONCLUSIONS

This paper tackles graph anomaly detection under limited supervision by introducing CONSISGAD. Specifically, our model leverages unlabeled data for consistency training and proposes a learnable data augmentation module for improved noise injection. Furthermore, we exploit the disparity in homophily distribution between normal and anomalous classes to build a tailored GNN backbone. Experiments demonstrate the superior performance of CONSISGAD over state-of-the-art methods.

ACKNOWLEDGMENTS

This research is supported by the National Research Foundation, Singapore under its AI Singapore Programme (AISG Award No: AISG2-TC-2021-002).

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

APPENDIX

## A  ADDITIONAL DETAILS OF TRAINING PARADIGM

We provide the pseudocode for training CONSISGAD in Algorithm 1. In Lines 2-3, we sample a batch of training nodes and a batch of unlabeled nodes for processing. In Lines 4-10, we construct a set of high-quality nodes from the unlabeled batch, and attach pesudo-labels to high-quality unlabeled nodes. In Lines 11-15, we train the learnable data augmentation based on the combination of label consistency and distribution diversity losses, during which the parameters of the graph encoder and predictor are frozen. In Lines 16-21, we train the graph encoder and the predictor with the combination of cross-entropy loss on labeled data and consistency training loss on high-quality unlabeled data, during which the parameters of the learnable augmentation are fixed.

---

**Algorithm 1** The Training Paradigm of CONSISGAD

**Input:** A graph $G = \{V, E, \boldsymbol{X}\}$, set of labeled nodes $V_{tr}$, set of unlabeled nodes $V_{un}$, graph encoder $g(\cdot; \theta_g)$, predictor $P(\cdot; \theta_p)$, learnable augmentation module $\Delta_{lm}(\cdot; \theta_{\Delta_{lm}})$, labeled batch size $B$, unlabeled batch size $\mu B$, anomalous threshold $\tau_a$, normal threshold $\tau_n$, weight of the label consistency loss $\alpha$.

**Output:** $\theta_g$ and $\theta_p$.

1: **while** not converged **do**
2:     Sample a batch of labeled nodes $\mathcal{X}_{tr} \subseteq V_{tr}$, with size $B$;
3:     Sample a batch of unlabeled nodes $\mathcal{X}_{un} \subseteq V_{un}$, with size $\mu B$;
4:     Initialize the set of high-quality nodes $\mathcal{X}_{hq} \leftarrow \emptyset$;
5:     **for** $v \in \mathcal{X}_{un}$ **do**
6:        $\boldsymbol{h}_v \leftarrow g(v; \theta_g)$;            ▷ *Calculate the embeddings.*
7:        $\boldsymbol{p}_v \leftarrow P(\boldsymbol{h}_v; \theta_p)$;          ▷ *Equation (1), conduct the prediction.*
8:        **if** $\boldsymbol{p}_v[0] \geq \tau_n$ or $\boldsymbol{p}_v[1] \geq \tau_a$ **then**
9:           $\tilde{\boldsymbol{y}}_v[\arg\max \boldsymbol{p}_v] \leftarrow 1$;       ▷ *Assign pesudo-label.*
10:          $\mathcal{X}_{hq} \leftarrow \mathcal{X}_{hq} \cup v$;          ▷ *Add to high-quality set.*
11:    **for** $v \in \mathcal{X}_{hq}$ **do**
12:       $\hat{\boldsymbol{h}}_v \leftarrow \Delta_{lm}(\boldsymbol{h}_v; \theta_{\Delta_{lm}})$;     ▷ *Equation (7), augmentation on high-quality nodes.*
13:       $\hat{\boldsymbol{p}}_v \leftarrow P(\hat{\boldsymbol{h}}_v; \theta_p)$;            ▷ *Equation (1), prediction.*
14:    $\mathcal{L}_{\Delta c} \leftarrow -\sum_{v \in \mathcal{X}_{hq}} \sum_{k=0}^{1} \tilde{\boldsymbol{y}}_v[k] \ln \hat{\boldsymbol{p}}_v[k]$,     $\mathcal{L}_{\Delta d} \leftarrow -\sum_{v \in \mathcal{X}_{hq}} d(\boldsymbol{h}_v, \hat{\boldsymbol{h}}_v)$;    ▷ *Equation (9)*
15:    Optimize $\alpha \mathcal{L}_{\Delta c} + \mathcal{L}_{\Delta d}$ while freezing $\theta_g$ and $\theta_p$;
16:    **for** $v \in \mathcal{X}_{tr}$ **do**
17:       $\boldsymbol{h}_v \leftarrow g(v; \theta_g)$;            ▷ *Calculate the embeddings.*
18:       $\boldsymbol{p}_v \leftarrow P(\boldsymbol{h}_v; \theta_p)$;          ▷ *Equation (1), prediction.*
19:    $\mathcal{L} \leftarrow -\sum_{v \in \mathcal{X}_{tr}} \sum_{k=0}^{1} \boldsymbol{y}_v[k] \ln \boldsymbol{p}_v[k]$; ▷ *Equation (2), cross-entropy loss on labeled nodes.*
20:    $\mathcal{L}_c \leftarrow -\sum_{v \in \mathcal{X}_{hq}} \sum_{k=0}^{1} \tilde{\boldsymbol{y}}_v[k] \ln \hat{\boldsymbol{p}}_v[k]$;        ▷ *Equation (3), consistency loss.*
21:    Optimize $\mathcal{L} + \mathcal{L}_c$ while freezing $\theta_{\Delta_{lm}}$;
22: **return** $\theta_g$ and $\theta_p$.

---

We provide the pseudocode for the sharpening function in Algorithm 2. In Line 1, we initialize an empty output vector with the same shape as the input vector for storing intermediate results. In Lines 3-4, we mask previously computed elements in the output vector. In Lines 5-6, we integrate the mask with the input vector, followed by a SoftMax operation to sharpen the result. In Line 7, we update the output vector and enter the next iteration. Finally, we return the output vector.

**Complexity analysis.** As illustrated in Algorithm 1, our model is composed of two primary components: consistency training and the training of the learnable data augmentation module. These components operate iteratively in each iteration of the process. This approach potentially incurs a higher computational cost compared to the standard training procedures of graph neural networks. In this part, we provide a complexity analysis and analyze the possibility of using it on large graphs.

At the outset, we focus on analyzing the complexity of our backbone GNN model, as outlined in Equation (4). This model is a critical component in Algorithm 1 for calculating node embeddings. Considering a target node $v$, in the first GNN layer, the function $\delta(\cdot, \cdot; \theta_\delta)$ is im-

---

**Algorithm 2** Sharpen Function

---

**Input** Input vector $\boldsymbol{h}$, drop ratio $\xi$, dimension of the input vector $d$, sharpening temperature $t$, small value to avoid logarithm of zero $\epsilon$.

**Output** Output vector $\hat{\boldsymbol{h}}$.

1: Initialize $\hat{\boldsymbol{h}} \leftarrow \boldsymbol{0}$           ▷ *Construct an empty vector with the same shape as the input.*
2: **for all** $i \in \{1, \ldots, \lfloor \xi d \rfloor\}$ **do**
3:      $\boldsymbol{m} \leftarrow (1 - \hat{\boldsymbol{h}})$
4:      $\hat{\boldsymbol{m}} \leftarrow \log(\boldsymbol{m} + \epsilon)$           ▷ *Mask Top-(i-1) elements.*
5:      $\boldsymbol{y} \leftarrow (-\boldsymbol{h} + \hat{\boldsymbol{m}})/t$
6:      $\hat{\boldsymbol{y}} \leftarrow \text{SoftMax}(\boldsymbol{y})$           ▷ *Integrate the mask with the input and sharpen the result.*
7:      $\hat{\boldsymbol{h}} \leftarrow \hat{\boldsymbol{h}} + \hat{\boldsymbol{y}} \cdot \boldsymbol{m}$           ▷ *Update the output vector*
8: **return** $1 - \hat{\boldsymbol{h}}$

---

plemented as $\text{MLP}(\boldsymbol{h}_v^{l-1} \| \boldsymbol{h}_u^{l-1})$. A typical example of this MLP is $\sigma(\boldsymbol{W}_\delta(\boldsymbol{h}_v^{l-1} \| \boldsymbol{h}_u^{l-1}) + \boldsymbol{b}_\delta)$. The computational complexity in this first layer is $O(2dd_X + \bar{N}d + 2d)$, where $d_X$ represents the dimension of the input feature vector, $d$ the intermediate dimension of the embeddings, and $\bar{N}$ the average node degree. Each subsequent layer contributes an additional complexity of $O(2d^2 + \bar{N}d + 2d)$. Given $L$ total GNN layers, the overall complexity of the GNN backbone sums up to $O(2dd_X + \bar{N}d + 2d + (L-1)(2d^2 + \bar{N}d + 2d)) = O(2dd_X + 2Ld^2 + 2Ld + L\bar{N}d - 2d^2)$.

In every iteration of Algorithm 1, we sample a batch of labeled and unlabeled nodes for subsequent computations. These batches are of sizes $B$ (for labeled nodes) and $\mu B$ (for unlabeled nodes), respectively. We can split the whole process into three steps as follows.

- We begin by selecting high-quality nodes from the sampled batch of unlabeled nodes. In Line 6, the GNN backbone introduces a complexity of $O(2dd_X + 2Ld^2 + 2Ld + L\bar{N}d - 2d^2)$, as previously analyzed. The prediction step in Line 7 has a complexity of $O(Kd + 2K)$, where $K$ is the number of classes. Lines 8-10 involve checking the high-quality nodes with a complexity of $K$. Overall, for a batch size of $B$, Lines 5-10 collectively result in a complexity of $O(\mu B(2dd_X + 2Ld^2 + 2Ld + L\bar{N}d - 2d^2 + Kd + 3K))$.

- Each high-quality node is augmented and predicted, forming the basis for the consistency and diversity loss calculations. For Line 12, the complexity of the Sharpen function in Algorithm 2 involves $O(8\lfloor \xi d \rfloor d + d)$ across its steps. Consequently, Line 12 in Algorithm 1 incurs a complexity of $O(8\lfloor \xi d \rfloor d + d^2 + 3d)$. Line 13, similar to Line 7, involves a complexity of $O(Kd + 2K)$. Line 14 includes forming both the consistency and diversity losses, with complexities of $O(\mu BK^2)$ and $O(2\mu Bd)$, respectively. Summing up, Lines 11-14 entail a complexity of $O(\mu B(8\lfloor \xi d \rfloor d + d^2 + 3d + Kd + 2K) + \mu BK^2 + 2\mu Bd)$.

- The consistency training involves complexity calculations similar to the earlier sections. Lines 17 and 18, as previously analyzed, involve a complexity of $O(2dd_X + 2Ld^2 + 2Ld + L\bar{N}d - 2d^2 + Kd + 2K)$. Lines 19 and 20, pertaining to the loss calculations, have complexities of $O(BK^2)$ and $O(\mu BK^2)$, respectively. Thus, the total complexity for Lines 16-20 over $B$ iterations is $O(B(2dd_X + 2Ld^2 + 2Ld + L\bar{N}d - 2d^2 + Kd + 2K) + BK^2 + \mu BK^2)$.

In each iteration of Algorithm 1, we aggregate the complexities from the previous sections to derive the overall complexity. This can be expressed as $O(2B(\mu + 1)dd_X + B(2\mu L + 2L - \mu - 2)d^2 + (2\mu BL + \mu BL\bar{N} + \mu BK + 8\mu B\lfloor \xi d \rfloor + 5\mu B + K\mu B + 2BL + BL\bar{N} + BK)d + 5\mu BK + 2\mu BK^2 + 2BK + BK^2)$.

Particularly in our anomaly detection scenario, where typically $K = 2$ due to the binary classification nature, and $L$ generally ranges from 1 to 3 as the number of GNN layers, the complexity of our model is predominantly influenced by the feature dimension $d_X$, the hidden dimension $d$, and the average node degree $\bar{N}$. This indicates that with appropriately set hyper-parameters, our model shows promising potential for application on large-scale graphs.

# B    DESCRIPTIONS OF DATASETS

The datasets are summarized in Table 3. The Amazon dataset (McAuley & Leskovec, 2013) aims to detect fraudsters paid to give fake reviews for products under the Musical Instruments category on Amazon.com. It includes three types of edges: U-P-U (users reviewing at least one same product), U-S-U (users having at least one same star rating within one week), U-V-U (users with top-5% mutual review TF-IDF similarities). The YelpChi dataset (Rayana & Akoglu, 2015) aims to identify anomalous hotel and restaurant reviews on Yelp.com. It contains three types of edges: R-U-R (reviews posted by the same user), R-S-R (reviews with the same star rating on the same product), R-T-R (reviews posted in the same month on the same product). The T-Finance dataset (Tang et al., 2022) aims to find anomalous accounts in a transaction network, including fraud, money laundering, and online gambling. The T-Social dataset (Tang et al., 2022) aims to catch abnormal users in a social network. Finally, the Industrial graph, sourced from Grab Holdings Inc., depicts a transaction graph in a real-world setting, capturing online transactions within a leading super app. To maintain anonymity, we omit the explicit details of the graph. However, it roughly comprises over a million nodes and tens of millions of edges.

Please note that limited supervision has been assessed in several state-of-the-art baselines, as indicated in (Tang et al., 2022). To ensure a fair comparison, we have employed the same data split as they did.

# C    DESCRIPTIONS OF THE BASELINES

In this section, we will introduce baselines used in more details, their hyper-parameter settings, and their implementation specifics.

## C.1    GENERIC GNN MODELS

**MLP (multi-layer perceptron (Rosenblatt, 1958)):** A multi-layer perceptron network with one hidden layer and ReLU activation.

**GCN (Graph Convolutional Network (Kipf & Welling, 2017)):** A graph neural network that performs graph spectral convolution via a localized first-order approximation of spectral filters.

**GraphSAGE (Graph Sample and AggregatE (Hamilton et al., 2017)):** A graph neural network that samples and aggregates neighboring features to generate node embeddings. It also proposes three ways of aggregation: mean, LSTM, and pooling. In the experiments, the mean aggregator is used.

**GAT (Graph Attention Networks (Veličković et al., 2018)):** A graph neural network that applies the attention mechanism to the neighborhood aggregation process. The number of attention heads is set to 2 in the experiments.

**GATv2 (Graph Attention Networks v2 (Brody et al., 2022)):** GAT_v2 improves GAT with a modified attention mechanism, which allows dynamic attention. The number of attention heads is set to 2 in our experiments.

**GIN (Graph Isomorphism Network (Xu et al., 2019)):** A graph neural network that generalizes the Weisfeiler-Lehman (WL) graph isomorphism test. In our experiments, the sum operation is used to aggregate neighboring features, and an MLP is deployed to update node embeddings. The deployed MLP consists of one hidden layer and ReLU activation.

The MLP is implemented via PyTorch (Paszke et al., 2019). For graph neural network models, we use the official implementation provided by DGL (Wang et al., 2019b).

## C.2    GAD MODELS

In this subsection, we will introduce the graph anomaly detection models employed in our experiments.

|  | # Nodes | # Edges | # Features | Anomaly | Train:Valid:Test |
|---|---|---|---|---|---|
| Amazon | 11,944 | 4,398,392 | 25 | 6.87% | 1%:33%:66% |
| YelpChi | 45,954 | 3,846,979 | 32 | 14.53% | 1%:33%:66% |
| T-Finance | 39,357 | 21,222,543 | 10 | 4.58% | 1%:33%:66% |
| T-Social | 5,781,065 | 73,105,508 | 10 | 3.01% | 0.01%:33.33%:66.66% |
| Industrial graph | ~1M | ~10M | 17 | < 0.6% | 1%:33%:66% |

Table 3: Statistical summary of the datasets used in our experiment.

**CARE-GNN (CAmouflage-REsistant Graph Neural Network (Dou et al., 2020))**[4]: This approach employs a neural classifier to estimate similarity between a node and its neighbors, and filters out dissimilar neighbors for a central node. The optimal filtering threshold is found through reinforcement learning. It also proposes a relation-aware neighbor aggregator to deal with different relation types in the graph.

**GraphConsis ((Liu et al., 2020))**[5]: This approach identifies three inconsistency issues in graph anomaly detection tasks, namely, context, feature, and relation inconsistencies. To deal with the context inconsistency, this work assigns each node a learnable context embedding to capture its local structure. To handle the feature inconsistency, this work filters neighbors based on estimated consistency scores from a neural classifier. To counter the relation inconsistency, this work trains an embedding for each relation and uses a self-attention mechanism to aggregate neighbors.

**PC-GNN (Pick and Choose Graph Neural Network (Liu et al., 2021b))**[6]: This approach adopts a label-balanced sampler to pick nodes and edges for training, where the sampling probability is inversely proportional to the label frequency. In addition, it proposes a neighborhood sampler that over-samples the neighborhood of fraud nodes and under-samples the neighborhood of normal ones.

**BWGNN (Beta Wavelet Graph Neural Network (Tang et al., 2022))**[7]: This approach finds that anomaly can cause energy shift from the low-frequency part to the high-frequency part in the graph spectral domain. It leverages the Beta kernel to create band-pass filters with spatial and spectral locality for anomaly detection. For multi-relation graphs, BWGNN provides two options, namely homo and hetero. The BWGNN(homo) converts the mutli-relation graph into a single graph for convolution, while the BWGNN (hetero) applies convolution to each relation separately and aggregates output from each relation via maximum pooling.

**H2-FDetector (Graph Neural Network-based Fraud Detector with Homophilic and Heterophilic Interactions (Shi et al., 2022))**[8]: This approach detects homophilic and heterophilic edges via an auxiliary neural classifier, which is trained with an additional loss function. It adopts a different aggregation strategy for detected heterophilic edges where the opposites of node embeddings are used. Additionally, it averages embeddings in each class and uses the averaged embedding as the class prototype. Node representations are required to be close to their corresponding class prototype.

**GHRN (Graph Heterophily Resistant Network (Gao et al., 2023a))**[9]: This approach theoretically proves that identifying heterophilic edges in the spatial domain is equivalent to extracting high-frequency signals in the spectral domain. It proposes to remove heterophilic edges based on graph spectral theory and to use the new graph for model training. As GHRN adopts BWGNN as its backbone, GHRN(homo) and GHRN(hetero) follow the same definition as BWGNN(homo) and BWGNN(hetero), respectively.

**GDN (Graph Decomposition Network (Gao et al., 2023b))**[10]: This approach partitions node representations into class features and surrounding features. For class features, it applies class constraints to ensure that nodes within the same class have similar class features. For surrounding fea-

---

[4]https://github.com/YingtongDou/CARE-GNN

[5]https://github.com/safe-graph/DGFraud-TF2

[6]https://github.com/PonderLY/PC-GNN

[7]https://github.com/squareRoot3/Rethinking-Anomaly-Detection

[8]https://github.com/shifengzhao/H2-FDetector

[9]https://github.com/blacksingular/GHRN

[10]https://github.com/blacksingular/wsdm$\_$GDN

tures, it applies connectivity constraints to ensure that neighboring nodes have similar surrounding features.

**GAGA (Group AGgregation enhanced TrAnsformer (Wang et al., 2023))**[11]: This approach explicitly uses partially observed labels to partition neighbors into three groups: normal, fraud, and unknown nodes. Each group of nodes is treated differently during processing. It augments node features with trainable hop, relation, and group embeddings, and uses a transformer encoder to transform node features.

# D    DETAILED EXPERIMENTAL SETTINGS

## D.1    DETAILS OF EVALUATION METRICS

We employ 3 metrics to systematically evaluate model performance. Following common settings in previous works (Dou et al., 2020; Tang et al., 2022), we select the Area Under the Receiver Operating Characteristic Curve (AUROC) as one of our metric. However, as indicated in (Davis & Goadrich, 2006), the AUC score sometimes gives an overly optimistic view of a model if the dataset has a highly skewed label distribution. To get a more thorough view of model performance, we pick another two metrics for evaluation: Area Under the Prevision Recall Curve (AUPRC) and F1-Macro, the unweighted mean of per-class F1-scores. To compute F1-Macro, we apply the threshold-moving strategy (Collell et al., 2018) to all baselines, which adjusts the classification threshold to achieve the best score in validation and directly uses the adjusted threshold in test. All metrics are in the range from 0 to 1, and a higher score indicates a better model performance.

## D.2    HYPER-PARAMETER SETTINGS

**Hyper-parameter settings of baselines.**    For the GAD baselines, we utilize the public code repositories provided by the authors, adhering to the default hyper-parameters specified in the original papers or the repositories. This ensures the integrity of our comparative analysis, as these parameters have been meticulously fine-tuned on the benchmark datasets by the authors to achieve optimal performance.

In particular, for CARE-GNN, the RL action step size is set as 0.02 and the similarity loss weight is set as 2. For GraphConsis, the number of layers is set as 2 while the sample numbers for the first and second layers are set to 10 and 5, respectively. For BWGNN, the order of the kernel function is set to 2 for Amazon, YelpChi, and T-Finance, and to 5 for T-Social. Two versions of the BWGNN are considered, namely the homo and hetero versions. For H2-FDetecto, the two hyper-parameters $\gamma_1$ and $\gamma_2$ are both set to 1.2 for YelpChi, T-Finance, and T-Social, while in Amazon, $\gamma_1$ and $\gamma_2$ are set to 0.4 and 1.4, respectively. For GHRN, the deleting ratio is set to 0.015 for Amazon, T-Finance, and T-Social, and to 0.1 for YelpChi. For GDN, the top-$K$ feature is set to 10 for all datasets. For GAGA, the number of hops and the number of heads in the transformer model are set to 2 and 4, respectively, for all datasets.

**Hyper-parameter settings of our CONSISGAD.**    For generic GNN baselines and our model, we set the dimension of hidden features as 64, number of layers as 1, the activation function as SeLU (Klambauer et al., 2017), and the number of epochs to be 100. Mini-batch training is adopted with a training batch size set to 32 for the Amazon dataset and to 128 for the others. Model parameters are optimized with the Adam optimizer (Kingma & Ba, 2015) where the learning rate is set to be 0.001 and weight decay to be 0.00001. When evaluating generic GNN models on multi-relation graphs, such as Amazon and YelpChi, we convert the graphs to corresponding single-relation graphs by merging multiple relation graphs together.

There are multiple important hyper-parameters in CONSISGAD, including the ratio of the unlabeled batch size to training batch size $\mu$[12], anomalous threshold $\tau_a$, normal threshold $\tau_n$, weight of the label consistency loss $\alpha$, distribution distance function $D(\cdot, \cdot)$, and the drop ratio $\xi$ in the sharpening function. Here we report the specific hyper-parameter values in each dataset in Table 4.

---

[11]https://github.com/Orion-wyc/GAGA

[12]In general, the unlabeled batch size is usually several times larger than the training batch size. Thus, this ratio represents this multiple.

| Methods | Amazon | YelpChi | T-Finance | T-Social |
|---|---|---|---|---|
| $\mu$ | 4 | 4 | 5 | 5 |
| $\tau_a$ | 88 | 85 | 82 | 88 |
| $\tau_n$ | 97 | 91 | 95 | 95 |
| $\alpha$ | 1.0 | 0.5 | 5.0 | 0.5 |
| $D(\cdot, \cdot)$ | euc. | cos. | euc. | euc. |
| $\xi$ | 0.3 | 0.1 | 0.2 | 0.1 |

Table 4: Hyper-parameter settings for CONSISGAD. Here, for space efficiency, we use *euc.* to stand for euclidean distance and *cos.* for cosine distance.

| Methods | Industrial Graph | | |
|---|---|---|---|
| | AUROC | AUPRC | Macro F1 |
| MLP | $99.35_{\pm0.11}$ | $39.35_{\pm3.09}$ | $75.83_{\pm0.16}$ |
| GCN | $82.70_{\pm0.11}$ | $8.38_{\pm0.64}$ | $62.58_{\pm0.55}$ |
| GraphSAGE | $99.64_{\pm0.02}$ | $62.44_{\pm0.51}$ | $83.18_{\pm0.10}$ |
| GAT | $85.94_{\pm0.77}$ | $12.83_{\pm1.67}$ | $65.31_{\pm0.98}$ |
| GIN | $94.74_{\pm0.33}$ | $44.53_{\pm1.91}$ | $73.78_{\pm0.68}$ |
| GATv2 | $88.85_{\pm0.35}$ | $14.50_{\pm0.65}$ | $65.94_{\pm0.50}$ |
| CARE-GNN | $99.56_{\pm0.02}$ | $43.74_{\pm1.18}$ | $73.27_{\pm1.16}$ |
| BWGNN(homo) | $98.46_{\pm0.50}$ | $62.37_{\pm2.75}$ | $80.80_{\pm0.60}$ |
| GHRN | $99.45_{\pm0.08}$ | $35.60_{\pm6.08}$ | $69.37_{\pm0.27}$ |
| GDN | $99.72_{\pm0.01}$ | $63.78_{\pm0.20}$ | $79.55_{\pm0.10}$ |
| GAGA | $\underline{99.75}_{\pm0.01}$ | $\underline{65.19}_{\pm0.79}$ | $\underline{82.37}_{\pm0.99}$ |
| CONSISGAD(GNN) | $99.74_{\pm0.01}$ | $67.26_{\pm0.49}$ | $82.73_{\pm0.33}$ |
| CONSISGAD | $\mathbf{99.77}_{\pm0.02}$ | $\mathbf{69.06}_{\pm0.63}$ | $\mathbf{83.10}_{\pm0.35}$ |

Table 5: Comparison (%) on Industrial Graph, with the best bolded and runner-up underlined.

## D.3 DATASETS

The data splitting is based on the stratified sampling in Scikit-learn (Pedregosa et al., 2011), which keeps a consistent anomaly ratio in all sets.

## D.4 EXPERIMENTAL ENVIRONMENT

All the experiments are conducted on a server running Ubuntu 22.04.2 with 3.10GHz Intel Xeon Gold 6346 CPU, 1024GB RAM, and 8 NVIDIA Tesla A100 GPUs with 80GB of memory each. CONSISGAD is implemented based on Python 3.7.15, PyTorch 1.13.1, and DGL 1.1.0.

# E ADDITIONAL EXPERIMENTS

## E.1 GRAPH ANOMALY DETECTION ON INDUSTRIAL GRAPH

Table 5 presents the experimental results obtained from the Industrial Graph, which is derived from a production environment. It is evident that CONSISGAD consistently achieves optimal performance across all three metrics, aligning with the observations made in our main paper. It is noteworthy that, given the pronounced imbalance characteristics inherent to real-world production settings, the AUROC is exceptionally high for all evaluated methods.

## E.2 PERFORMANCE UNDER VARYING SUPERVISION

Figure 6 and Figure 7 illustrate the performance of the model under varying levels of supervision, as measured by the AUROC and Macro F1 metrics, respectively. The observations align with the discussion presented in the main paper.

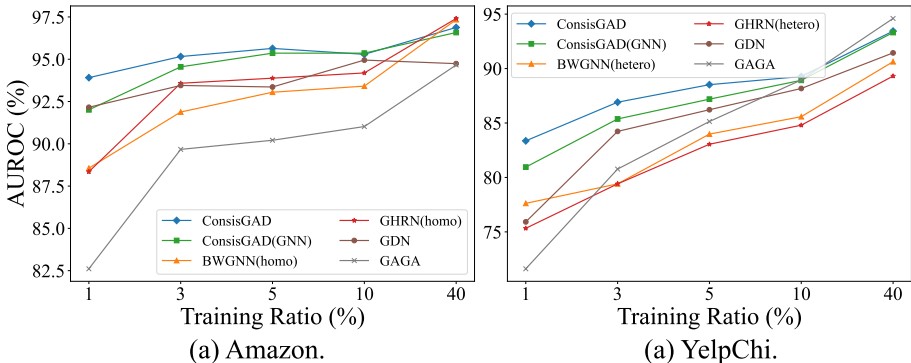

Figure 6: Experiments under varied supervision in terms of the AUROC metric.

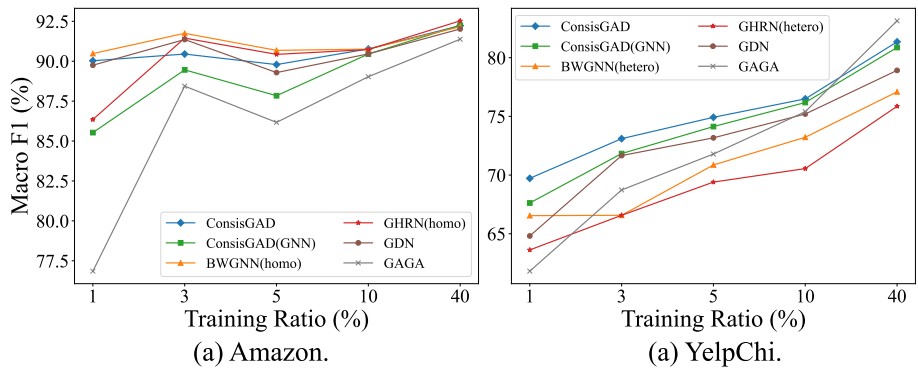

Figure 7: Experiments under varied supervision in terms of the Macro F1 metric.

### E.3 Influence of Label Consistency and Distribution Diversity

Figure 8(a) and Figure 8(b) visualize the contributions of the label consistency and distribution diversity to CONSISGAD, in terms of AUROC and Macro F1 scores, respectively. We can achieve a similar conclusion as discussed in Section 4.2.2.

### E.4 Comparison with Other Graph Augmentation Techniques

In this section, we benchmark our learnable data augmentation against several stochastic augmentation methods. Specifically, we substitute our augmentation module with widely-recognized techniques in the consistency training framework, including DropNode (Feng et al., 2020a), DropEdge (Rong et al., 2020), and Dropout (Srivastava et al., 2014). We apply Dropout to both input features, denoted as DropInput, and intermediate features, referred to as DropHidden. It is noteworthy that DropHidden serves as a non-learnable analogue to our augmentation module. For a fair comparison, we determine the optimal drop rate for each method within the interval of $(0, 0.5]$, with a step size of $0.1$. The outcomes, depicted in Figure 9 through the three metrics, underscore the preeminence of our proposed learnable data augmentation over the traditional stochastic alternatives. A salient observation is that among conventional augmentation methods, employing DropHidden and DropEdge in consistency training usually yield more substantial and stable performance improvements across all datasets. This empirical insight substantiates our decision to integrate learnable data augmentation at intermediate states and suggests the potential exploration of applying such augmentation to graph topology (Xia et al., 2022). We also list the optimal drop rate in Table 6.

### E.5 Flexibility with other GNN models

We investigate how traditional GNN models perform when equipped with consistency training and learnable data augmentation. We pick two representative GNN models, namely GCN and Graph-

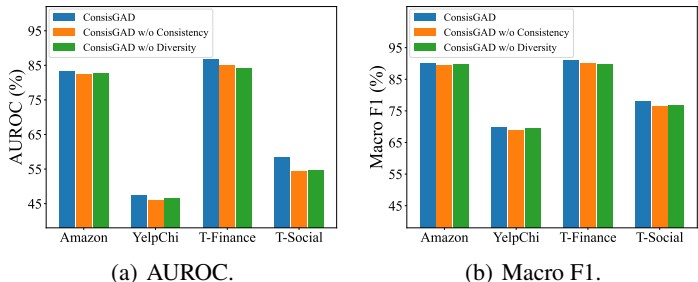

(a) AUROC.         (b) Macro F1.

Figure 8: Influence of Label Consistency and Distribution Diversity. Subfigure (a) depicts the result on the AUROC metric. Subfigure (b) depicts the result on the Macro F1 metric.

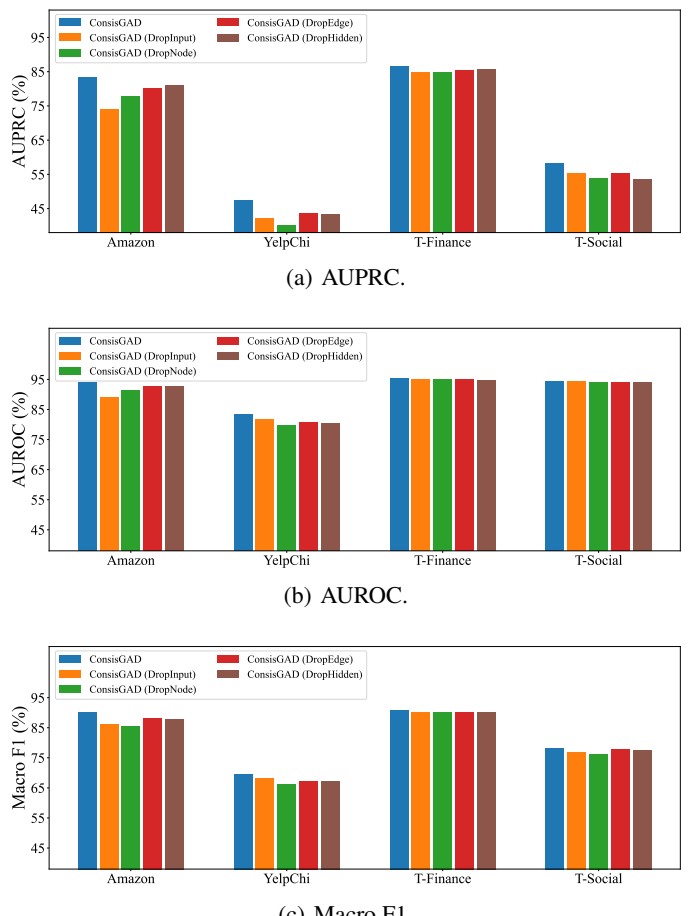

(a) AUPRC.

(b) AUROC.

(c) Macro F1.

Figure 9: Comparison between the learnable augmentation and traditional stochastic augmentation. Subfigure (a) depicts the result on the AUPRC metric. Subfigure (b) depicts the result on the AUROC metric. Subfigure (c) depicts the result on the Macro F1 metric.

SAGE, and replace our backbone model with them. As indicated in Tables 7, GNN model trained with consistency training and learnable data augmentation consistently outperforms its counterpart across all metrics, which showcases the flexibility of CONSISGAD to facilitate the training of various base GNN architectures.

| Methods | Amazon | YelpChi | T-Finance | T-Social |
|---|---|---|---|---|
| CONSISGAD(DropInput) | 0.1 | 0.1 | 0.1 | 0.4 |
| CONSISGAD(DropNode) | 0.2 | 0.1 | 0.1 | 0.5 |
| CONSISGAD(DropEdge) | 0.2 | 0.1 | 0.1 | 0.3 |
| CONSISGAD(DropHidden) | 0.3 | 0.1 | 0.2 | 0.1 |

Table 6: The optimal drop rate for each augmentation technique in each dataset.

| Methods | Amazon | | | YelpChi | | | T-Finance | | |
|---|---|---|---|---|---|---|---|---|---|
| | AUROC | AUPRC | Macro F1 | AUROC | AUPRC | Macro F1 | AUROC | AUPRC | Macro F1 |
| GCN | $87.34_{\pm 0.59}$ | $48.06_{\pm 2.73}$ | $70.94_{\pm 2.43}$ | $54.65_{\pm 0.53}$ | $17.07_{\pm 0.44}$ | $35.59_{\pm 10.27}$ | $\mathbf{89.29}_{\pm 0.19}$ | $53.94_{\pm 3.22}$ | $77.16_{\pm 1.20}$ |
| w CONSISGAD | $\mathbf{87.72}_{\pm 0.42}$ | $\mathbf{48.42}_{\pm 1.56}$ | $\mathbf{73.88}_{\pm 1.42}$ | $\mathbf{54.96}_{\pm 0.54}$ | $\mathbf{17.40}_{\pm 0.55}$ | $\mathbf{43.06}_{\pm 7.40}$ | $89.28_{\pm 0.22}$ | $\mathbf{57.41}_{\pm 2.05}$ | $\mathbf{78.19}_{\pm 0.84}$ |
| GraphSAGE | $90.12_{\pm 0.48}$ | $73.17_{\pm 4.65}$ | $84.25_{\pm 2.26}$ | $73.70_{\pm 0.52}$ | $34.57_{\pm 0.78}$ | $63.33_{\pm 0.51}$ | $89.42_{\pm 1.36}$ | $49.08_{\pm 6.34}$ | $77.62_{\pm 1.87}$ |
| w CONSISGAD | $\mathbf{91.94}_{\pm 0.38}$ | $\mathbf{79.61}_{\pm 1.69}$ | $\mathbf{88.78}_{\pm 0.61}$ | $\mathbf{73.82}_{\pm 0.47}$ | $\mathbf{34.88}_{\pm 0.58}$ | $\mathbf{63.41}_{\pm 0.2}$ | $\mathbf{91.23}_{\pm 0.80}$ | $\mathbf{56.85}_{\pm 3.28}$ | $\mathbf{80.90}_{\pm 1.01}$ |

Table 7: The performance (%) of traditional GNN models when equipped with CONSISGAD.

### E.6 ANALYSIS OF THE QUALITY OF HIGH-QUALITY NODES

In this subsection, we examine the performance of high-quality nodes. We visualize Macro F1 scores of high-quality nodes at each epoch during training and compare these with scores from the test set. Figure 10 depicts the dynamic of Macro F1 scores for both high-quality and test nodes. Our findings reveal that the high-quality nodes consistently exhibit higher Macro F1 scores compared to test nodes. Specifically, the average performance on high-quality nodes surpasses that on test nodes by margins of 1.01%, 0.83%, 0.71%, and 0.64% on the Amazon, YelpChi, T-Finance, and T-Social datasets, respectively. Generally, both high-quality nodes and test nodes show a gradual increase in performance, with the former driving the improvement of the latter. An exception lies in the T-Social dataset, on which the performance fluctuates a bit. The underlying reason might be that the batch size and training data size are relatively small compared to the size of the whole dataset, which causes fluctuation of the performance. This outcome highlights the superior quality of high-quality nodes and validates the effectiveness of our selection criteria for consistency training.

### E.7 ANALYSIS OF HYPER-PARAMETER SENSITIVITY

**Weight of the label consistency loss $\alpha$.** Figure 11 illustrates the effect of different label consistency weights $\alpha$ on model performance. We experiment with a range of $\alpha$ values: {0.1, 0.2, 0.5, 1.0, 2.0, 5.0, 10.0}, and measure the corresponding model performance. Our result indicates that an $\alpha$ value of around 1.0 yields optimal performance for the Amazon and YelpChi datasets, whereas a value of around 5.0 is preferable for the T-Finance dataset. Overall, model performance remains stable with a moderate $\alpha$ value. If $\alpha$ is set to a excessively high value, the performance will be greatly impaired. This is likely due to the diminished diversity in generated augmentations, which is crucial for effective consistency training.

**Drop ratio $\xi$.** Figure 12 depicts the impact of various drop ratios $\xi$ on model performance. In this analysis, We test drop ratios $\xi$ in the range of {0.0, 0.1, 0.2, 0.3, 0.4, 0.5}. The result reveals that maintaining $\xi$ within 0.1 to 0.3 is a foolproof decision for all datasets. If $\xi$ is set too small, CONSISGAD degrades to the backbone model and cannot leverage abundant information encoded in unlabeled data for training. Conversely, a higher $\xi$ risks losing important information, making it hard for the model to discern valuable patterns.

**Normal threshold $\tau_n$ and anomalous threshold $\tau_a$.** Figure 13 and Figure 14 illustrate how the model behaves under different normal thresholds $\tau_n$ and anomalous thresholds $\tau_a$, respectively. We experiment with normal thresholds $\tau_n$ set at {89, 91, 93, 95, 97, 99} and anomalous thresholds $\tau_a$ at {79, 92, 85, 88, 91}. Overall, the result suggests that our model, CONSISGAD, exhibits robustness against the range of tested thresholds. This resilience likely stems from the adaptability of our learnable augmentation module, which can dynamically adjust to varying threshold values during training.

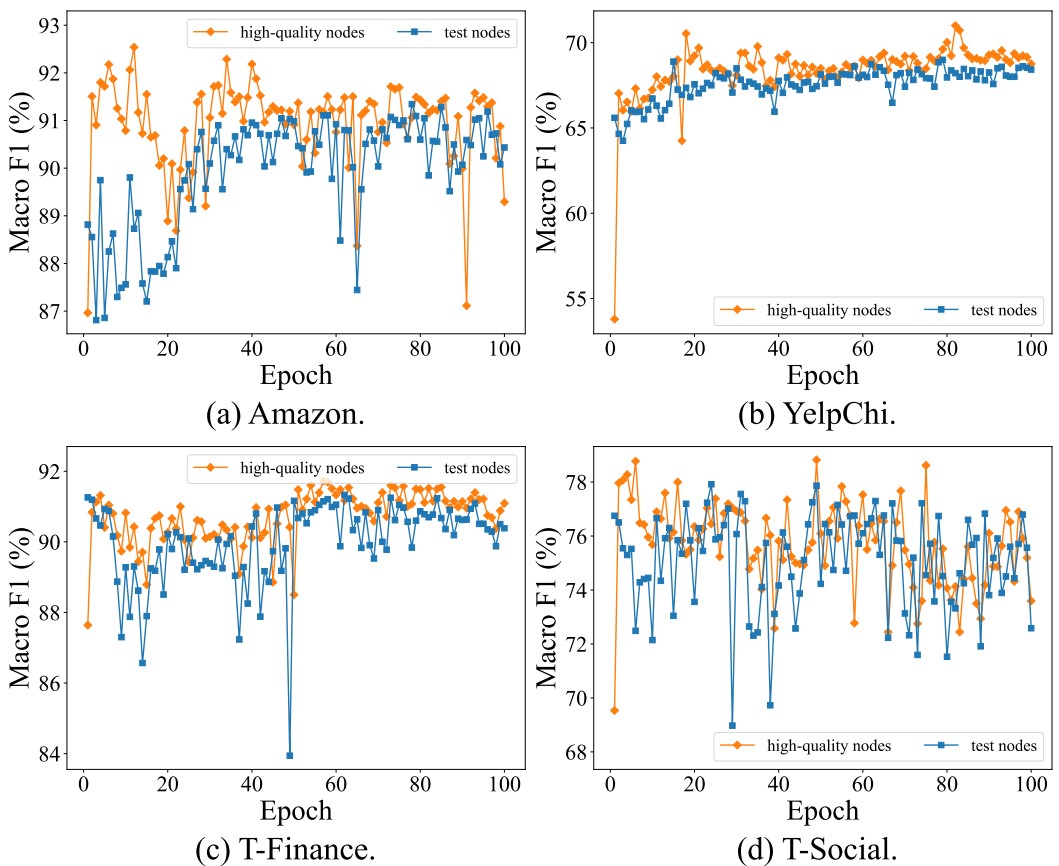

Figure 10: Dynamic of Macro F1 scores on high-quality nodes and test nodes. The X-axis denotes epochs during training and Y-axis the corresponding Macro F1 score. The orange lines represents the high-quality nodes, while the blue lines represent the test nodes. Subfigre(a)-(d) depict the result on Amazon, YelpChi, T-Finance, and T-Social, respectively.

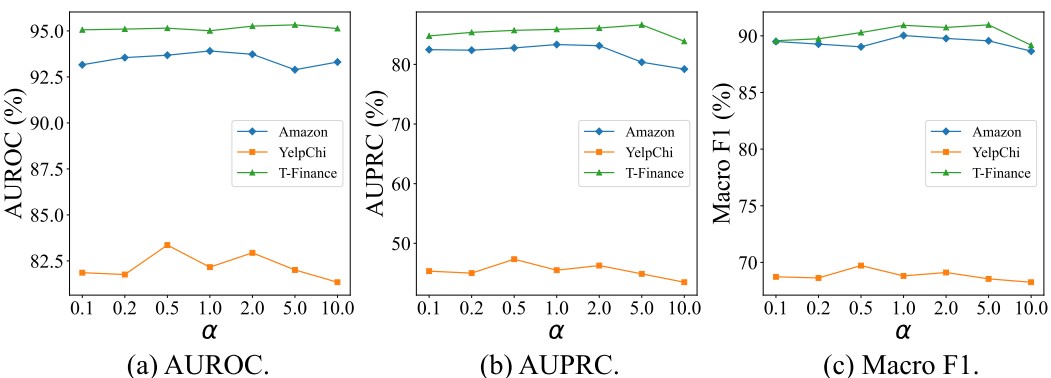

Figure 11: The performance of CONSISGAD with varied weights of the label consistency loss $\alpha$ in terms of AUROC (Subfigure a), AUPRC (Subfigure b), and Macro F1 (Subfigure c). Blue, orange, and green lines depict results on Amazon, YelpChi, and T-Finance, respectively.

## E.8 PERFORMANCE OF THE BACKBONE GNN MODEL ON MORE GRAPHS.

In this subsection, we examine the performance of our backbone GNN model on generic multi-class node classification tasks with diverse homophily ratios. We follow the experimental procedure of a recent work, the Feature Selection Graph Neural Network (FSGNN) (Maurya et al., 2022), and use

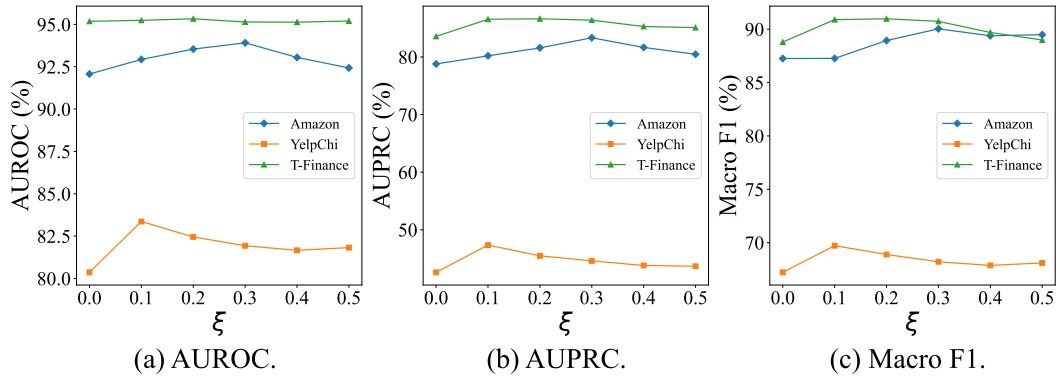

(a) AUROC.                 (b) AUPRC.                 (c) Macro F1.

Figure 12: The performance of CONSISGAD with varied drop ratios $\xi$ in terms of AUROC (Subfigure a), AUPRC (Subfigure b), and Macro F1 (Subfigure c). Blue, orange, and green lines depict results on Amazon, YelpChi, and T-Finance, respectively.

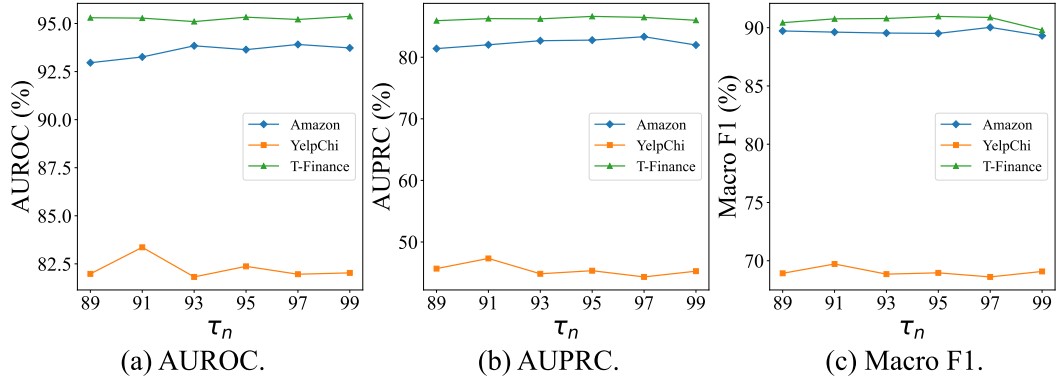

(a) AUROC.                 (b) AUPRC.                 (c) Macro F1.

Figure 13: The performance of CONSISGAD with varied normal thresholds $\tau_n$ in terms of AUROC (Subfigure a), AUPRC (Subfigure b), and Macro F1 (Subfigure c). Blue, orange, and green lines depict results on Amazon, YelpChi, and T-Finance, respectively.

their public repository [13] for experimentation. The experiment consists of three homophily graphs and six heterophily graphs, with a homophily ratio ranging from 0.11 to 0.81. For each dataset, we compute the average accuracy across ten publicly available data splits for evaluation, which are widely adopted by researchers in the community (Pei et al., 2020; Zhu et al., 2020; Maurya et al., 2022). For our GNN backbone model, we set the number of layers to be two and keep the remaining architecture untouched. For each dataset, we carry out a lightweight fine-tuning on the learning rate and weight decay. Specifically, we pick the best learning rate from {0.1, 0.01, 0.001} and weight decay from {0.01, 0.001, 0.0001} based on the validation performance. Table 8 summarizes dataset statistics and corresponding results, where the performance of other baselines are taken from (Maurya et al., 2022).

Note that our proposed GNN backbone, CONSISGAD(GNN), is based the difference of contextual homophily distribution between normal and anomalous nodes. Such a phenomenon is commonly seen in the graph anomaly detection task (i.e., the imbalanced binary classification task) where normal nodes have high homophily distribution while anomalous nodes have low homophily distribution. This prominent distribution discrepancy lays down the foundation of our GNN backbone. When it comes to multi-class node classification tasks, the homophily-aware neighborhood aggregation (Equation (4)) in the GNN backbone will model the label distribution within a node's neighborhood. If this neighborhood label distribution exhibits distinguishable patterns for different classes, we would expect our backbone model to function well. From Table 8, we observe that our GNN backbone model performs comparably to existing baselines in the three homophily graphs.

---

[13]https://github.com/sunilkmaurya/FSGNN/tree/main

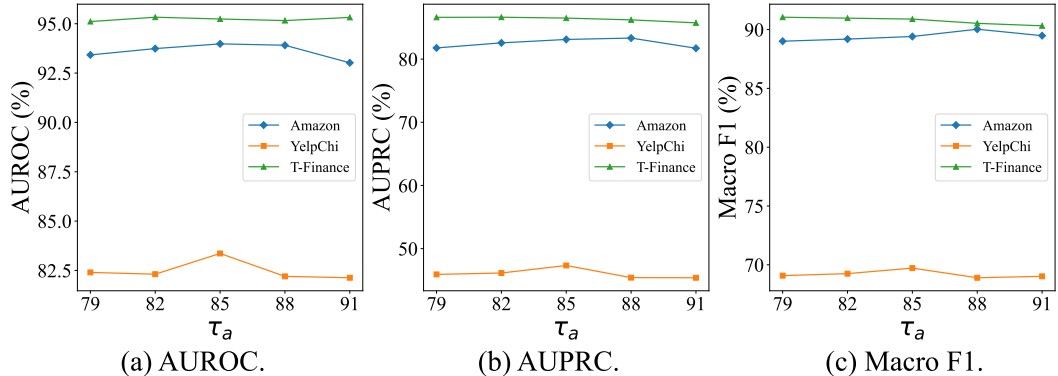

(a) AUROC.      (b) AUPRC.      (c) Macro F1.

Figure 14: The performance of CONSISGAD with varied anomalous thresholds $\tau_a$ in terms of AUROC (Subfigure a), AUPRC (Subfigure b), and Macro F1 (Subfigure c). Blue, orange, and green lines depict results on Amazon, YelpChi, and T-Finance, respectively.

| | Cora | Citeseer | Pubmed | Chameleon | Wisconsin | Texas | Cornell | Squirrel | Actor | Mean Acc. |
|---|---|---|---|---|---|---|---|---|---|---|
| Hom. ratio | 0.81 | 0.74 | 0.80 | 0.23 | 0.21 | 0.11 | 0.30 | 0.22 | 0.22 | |
| # Nodes | 2,708 | 3,327 | 19,717 | 2,277 | 251 | 183 | 183 | 5,201 | 7,600 | |
| # Edges | 5,278 | 4,732 | 44,338 | 36,101 | 499 | 309 | 295 | 198,353 | 26,659 | |
| # Features | 1,433 | 3,703 | 500 | 2,325 | 1,703 | 1,703 | 1,703 | 2,089 | 932 | |
| # Classes | 7 | 6 | 3 | 5 | 5 | 5 | 5 | 5 | 5 | |
| GCN | $87.28_{\pm1.26}$ | $76.68_{\pm1.64}$ | $87.38_{\pm0.66}$ | $59.82_{\pm2.58}$ | $59.80_{\pm6.99}$ | $59.46_{\pm5.25}$ | $57.03_{\pm4.67}$ | $36.89_{\pm1.34}$ | $30.26_{\pm0.79}$ | 61.62 |
| GAT | $82.68_{\pm1.80}$ | $75.46_{\pm1.72}$ | $84.68_{\pm0.44}$ | $54.69_{\pm1.95}$ | $55.29_{\pm8.71}$ | $58.38_{\pm4.45}$ | $58.92_{\pm3.32}$ | $30.62_{\pm2.11}$ | $26.28_{\pm1.73}$ | 58.55 |
| GraphSAGE | $86.90_{\pm1.04}$ | $76.04_{\pm1.30}$ | $88.45_{\pm0.50}$ | $58.73_{\pm1.68}$ | $81.18_{\pm5.56}$ | $82.43_{\pm6.14}$ | $75.95_{\pm5.01}$ | $41.61_{\pm0.74}$ | $34.23_{\pm0.99}$ | 69.50 |
| Cheby+JK | $85.49_{\pm1.27}$ | $74.98_{\pm1.18}$ | $89.07_{\pm0.30}$ | $63.79_{\pm2.27}$ | $82.55_{\pm4.57}$ | $78.38_{\pm6.37}$ | $74.59_{\pm7.87}$ | $45.03_{\pm1.73}$ | $35.14_{\pm1.37}$ | 69.89 |
| MixHop | $87.61_{\pm0.85}$ | $76.26_{\pm1.33}$ | $85.31_{\pm0.61}$ | $60.50_{\pm2.53}$ | $75.88_{\pm4.90}$ | $77.84_{\pm7.73}$ | $73.51_{\pm6.34}$ | $43.80_{\pm1.48}$ | $32.22_{\pm2.34}$ | 68.10 |
| GEOM-GCN | 85.27 | **77.99** | 90.05 | 60.90 | 64.12 | 67.57 | 60.81 | 38.14 | 31.63 | 64.05 |
| GCNII | $88.01_{\pm1.33}$ | $77.13_{\pm1.38}$ | **$90.30_{\pm0.37}$** | $62.48_{\pm2.74}$ | $81.57_{\pm4.98}$ | $77.84_{\pm5.64}$ | $76.49_{\pm4.37}$ | N/A | N/A | - |
| H2GCN-1 | $86.92_{\pm1.37}$ | $77.07_{\pm1.64}$ | $89.40_{\pm0.34}$ | $57.11_{\pm1.58}$ | $86.67_{\pm4.69}$ | $84.86_{\pm6.77}$ | $82.16_{\pm4.80}$ | $36.42_{\pm1.89}$ | $35.86_{\pm1.03}$ | 70.71 |
| WRGAT | $88.20_{\pm2.26}$ | $76.81_{\pm1.89}$ | $88.52_{\pm0.92}$ | $65.24_{\pm0.87}$ | $86.98_{\pm3.78}$ | $83.62_{\pm5.50}$ | $81.62_{\pm3.90}$ | $48.85_{\pm0.78}$ | $36.53_{\pm0.77}$ | 72.93 |
| GPRGNN | **$88.49_{\pm0.95}$** | $77.08_{\pm1.63}$ | $88.99_{\pm0.40}$ | $66.47_{\pm2.47}$ | $85.88_{\pm3.70}$ | $86.49_{\pm4.83}$ | $81.89_{\pm6.17}$ | $49.03_{\pm1.28}$ | $36.04_{\pm0.96}$ | 73.37 |
| FSGNN | $88.23_{\pm1.17}$ | $77.40_{\pm1.93}$ | $89.78_{\pm0.38}$ | **$78.95_{\pm0.86}$** | **$88.43_{\pm3.22}$** | **$87.57_{\pm4.86}$** | **$87.84_{\pm6.19}$** | **$74.10_{\pm1.89}$** | $35.75_{\pm0.96}$ | **78.67** |
| CONSISGAD(GNN) | $86.32_{\pm1.72}$ | $75.83_{\pm1.81}$ | $89.39_{\pm0.34}$ | $44.82_{\pm2.96}$ | $86.47_{\pm4.51}$ | $83.24_{\pm5.77}$ | $83.51_{\pm6.89}$ | $33.08_{\pm0.79}$ | **$37.38_{\pm1.55}$** | 68.89 |

Table 8: Mean classification accuracy on generic multi-class node classification tasks with different homophily ratios. Results of baseline models are taken from (Maurya et al., 2022). For FSGNN, we list the performance of the best variant for each dataset. Best results are bolded and runner-up underlined. Here, "N/A" denotes non-reported results.

Our GNN backbone is built upon the Message-Passing Neural Network (MPNN) framework, which allows handling homophily information naturally. For heterophily graphs, our backbone model achieves good performance in the Wisconsin, Texas, and Cornell datasets, but faces challenges in the Chameleon, Squirrel, and Actor datasets. Our further investigations in Figure 15 reveal that, in Wisconsin, Texas, and Cornell, nodes of different classes have distinct label distribution among their neighborhood, which allows our backbone model to distinguish different classes effectively. However, such distinct patterns are absent in Chameleon, Squirrel, and Actor, explaining the struggle of our backbone model on these datasets. Notably, despite a low accuracy on the Actor dataset, our model achieves a new state-of-the-art result, indicating its potential in handling heterophily graphs in multi-class node classification tasks. Future work may explore integrating our GNN backbone with other techniques to enhance performance on heterophily graphs.

### E.9  INFLUENCE OF EXACT LABELS FOR CONSISTENCY TRAINING

In our current settings, both labeled and unlabeled data are used for consistency training, but labeled samples are treated as unlabeled, using their predicted labels instead of exact labels. In this subsection, we investigate the effectiveness of using exact labels for labeled nodes during consistency training. Our results, presented in Table 9 and Table 10, suggest that using exact labels in consistency training does not markedly enhance performance across most datasets. This lack of improvement may stem from the fact that labeled data has already been utilized in the cross-entropy loss, and their reuse in the consistency training does not contribute significantly to performance enhancement. Notably, a decrease in performance is observed in the T-Social dataset. We hypothesize

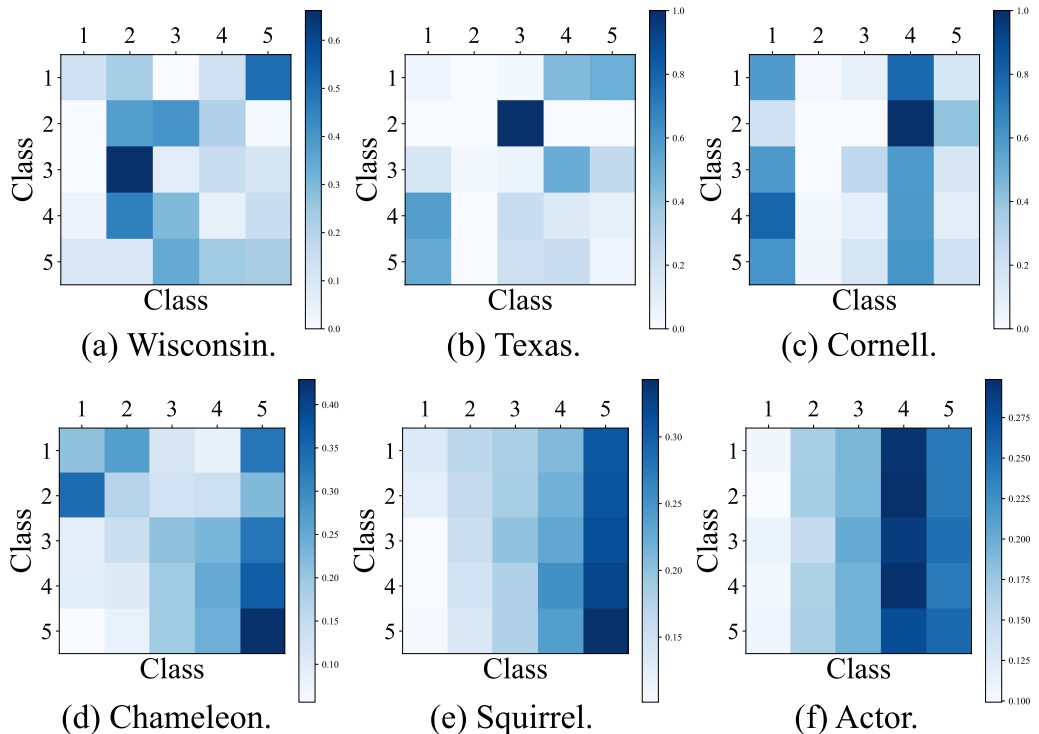

Figure 15: Neighborhood label distribution for different classes of nodes. Each row represents node class, while each column represents neighbor class. A darker color denotes a higher probability. The difference between two rows depicts the difference of neighborhood label distribution between corresponding classes. Rows in Subfigure (a)-(c) are distinguishable. Therefore, Subfigure (a)-(c) represent the set of graphs where nodes of different classes have distinct label distribution among their neighborhood, which are suitable to CONSISGAD(GNN). On the other hand, rows in Subfigure (d)-(f) are less distinguishable, which poses challenges to our backbone model.

| Methods | Amazon | | | YelpChi | | |
|---|---|---|---|---|---|---|
| | AUROC | AUPRC | Macro F1 | AUROC | AUPRC | Macro F1 |
| CONSISGAD | $93.91_{\pm0.58}$ | $83.33_{\pm0.34}$ | $90.03_{\pm0.53}$ | $83.36_{\pm0.53}$ | $47.33_{\pm0.58}$ | $69.72_{\pm0.30}$ |
| CONSISGAD (Exact Label) | $93.82_{\pm0.55}$ | $83.16_{\pm0.46}$ | $89.88_{\pm0.30}$ | $82.91_{\pm0.36}$ | $47.23_{\pm0.98}$ | $69.44_{\pm0.29}$ |

Table 9: Model performance when using exact labels of labeled nodes in consistency training on Amazon and YelpChi datasets.

| Methods | T-Finance | | | T-Social | | |
|---|---|---|---|---|---|---|
| | AUROC | AUPRC | Macro F1 | AUROC | AUPRC | Macro F1 |
| CONSISGAD | $95.33_{\pm0.30}$ | $86.63_{\pm0.44}$ | $90.97_{\pm0.63}$ | $94.31_{\pm0.20}$ | $58.38_{\pm2.10}$ | $78.08_{\pm0.54}$ |
| CONSISGAD (Exact Label) | $95.09_{\pm0.27}$ | $86.94_{\pm0.13}$ | $91.32_{\pm0.18}$ | $93.87_{\pm0.30}$ | $56.54_{\pm1.86}$ | $77.42_{\pm0.67}$ |

Table 10: Model performance when using exact labels of labeled nodes in consistency training on T-Finance and T-Social datasets.

this could be attributed to the model overfitting to label noise, which could be mitigated by the usage of predicted labels.

# F    DISCUSSION OF DATA AUGMENTATION ON GRAPHS

## F.1    DATA AUGMENTATION ON GRAPHS

Graph data augmentation techniques can be classified into four categories. Stochastic augmentation randomly samples noise to node features and graph structures (Srivastava et al., 2014; Feng et al., 2020a; Bo et al., 2022). Adversarial perturbation (Deng et al., 2019; Feng et al., 2019; Kong et al., 2022) noises node features by calculating virtual adversarial perturbations. Generative-model based augmentation employs generative models to generate artificial features for augmentation (Zhang et al., 2019; Zhao et al., 2021; Liu et al., 2022). Interpolation-based augmentation utilizes the MixUp-like (Zhang et al., 2018; Verma et al., 2019) methods to synthetize instances on graphs (Verma et al., 2021; Wang et al., 2021). Nonetheless, these approaches fail to calibrate the extent of data augmentation and suffer from over- or under-augmentation. In the realm of graph contrastive learning, several automatic data augmentation approaches have been proposed to address these limitations (You et al., 2021; Zhu et al., 2021), and we provide an in-depth comparison between our learnable augmentation and these methods in Appendix F.2.

## F.2    DETAILED COMPARISON WITH EXISTING AUTOMATIC DATA AUGMENTATION TECHNIQUES

In the realm of graph contrastive learning, various automatic data augmentation methods, such as JOAO (You et al., 2021) and GCA (Zhu et al., 2021), have been introduced. JOAO learns a sampling distribution for augmentation pairs using a min-max optimization process, while GCA generates augmented graphs by adaptively dropping edges and node features based on node centrality scores. Our learnable augmentation module presents notable differences from these approaches, particularly in terms of augmentation quality evaluation and learning objectives.

**Augmentation quality evaluation.**    JOAO adopts an adversarial training strategy, prioritizing augmentations that yield the greatest contrastive loss. This approach inherently aims to enhance the distribution diversity of augmentations. However, it may inadvertently introduce excessive noise due to overlooked label consistency, leading to suboptimal performance, as discussed in prior studies (Balaji et al., 2019; Tsipras et al., 2018). GCA assumes that a good augmentation should drop unimportant edges and features while keeping important ones. The importance is measured through node centrality scores. This method, while intuitive, does not necessarily guarantee high consistency and diversity in augmentations. In contrast, our work does not rely on pre-defined node importance metrics. Instead, we posit that effective augmentations should enhance a node's diversity without altering its label, maintaining high label consistency. To this end, we formulate differentiable distribution diversity and label consistency metrics to evaluate augmentation quality and to guide the training of our augmentation module.

**Learning objectives.**    The objective of JOAO is to select the optimal augmentation pair from a predetermined pool, relying heavily on domain knowledge for pool construction and configuration. The adaptive augmentation module in GCA remains fixed and non-learnable throughout the training process. This is due to the static nature of node centrality scores, resulting in constant dropping rates for edges and node features. Furthermore, it operates independently of the specific GNN encoder employed. By contrast, our work introduces a novel learnable augmentation module through learnable masking. This module is designed to synthesize custom augmentations for individual nodes, updating continuously throughout training to adapt to the dataset and evolving GNN encoder.

