# OpenReview forum: "Consistency Training with Learnable Data Augmentation for Graph Anomaly Detection with Limited Supervision"
_ICLR.cc/2024/Conference — ICLR 2024 spotlight_

### Official Review · Reviewer_tbTo · 2023-10-28

**Soundness:** 3 good
**Presentation:** 3 good
**Contribution:** 3 good
**Rating:** 5
**Confidence:** 4

**Summary:**

This paper has addressed the challenge of graph anomaly detection under limited supervision by introducing a novel model, which effectively leverages the abundance of unlabeled data for consistency training by incorporating a novel learnable data augmentation mechanism. Furthermore, CONSISGAD exploits the disparity in homophily distribution between normal and anomalous nodes, enabling the formulation of a refined GNN backbone that enhances its discriminatory power between the two classes.

**Strengths:**

1. The logic of the whole paper is relatively clear and there are no obvious grammatical errors.

2. There are quite sufficient experiments to verify the proposed method.

**Weaknesses:**

1) why Eq4 can achieve the edge-level homophily representation? where does it differ from GCN aggregation?

2) as you said, GAD has low homophily distribution, whether  the homophily-aware neighbourhood aggregation you propose will be effective for abnormal nodes, I deeply doubt it.

3) I don't find  the definition of Vhq in Section 2,  whiat is the meaning of Vun?

4) please compare  your learnable augmentation with other augmentation methods

5) please give complexity analysis

**Questions:**

see above

---

> ### Author Response · Authors · 2023-11-18
>
> Dear reviewer tbTo,
> Thank you for your thoughtful and constructive feedback on our submission 9315. We appreciate the time you have taken to review our work and provide detailed insights. Below, we address each of the concerns and questions you raised.
>
> **1. Why Eq4 can achieve the edge-level homophily representation? Where does it differ from GCN aggregation?**
>
> *For the first question: Why Eq4 can achieve the edge-level homophily representation?*
>
> In Eq.(4), the representation of an edge (e.g., $(v,u)$) is calculated using the representations of its two end nodes (e.g., $v$ and $u$) through the function $\delta(\cdot,\cdot;\theta_{\delta})$. Furthermore, Eq.(4) aggregates these edge-level representations into the node representation of node $v$. Notably, the node representation ($\mathbf{h}^l_v$) is calculated based on the distinctions in homophily distribution between normal and anomalous nodes, as discussed in the Introduction.
>
> The cross-entropy loss function elicits inverse predictions for normal and anomalous nodes. Specifically, this loss function steers the edge representations of edges surrounding a normal node away from those surrounding an anomalous node, emphasizing their distinct homophily distributions. As a result, edges with high homophily around normal nodes share similar representations, while edges with low homophily around anomalous nodes exhibit analogous representations. Therefore, through this optimization process, Eq.(4) achieves disparate representations for high-homophily and low-homophily (high-heterophily) edges, ensuring edge-level homophily representations for edges around different nodes.
>
> To validate the ability of the edge representations to capture edge-level homophily distribution, we conducted an analysis and visualized the results in Figure 3. The visualizations demonstrate that edges with the same homophily or heterophily labels can be well-clustered together. This indicates that the edge-level homophily representation adeptly reflects the type of edge homophily, forming a robust foundation for aggregation to represent the contextual homophily distribution of each node.
>
> *For the second question: Where does it differ from GCN aggregation?*
>
> Our backbone GNN model, as outlined in Eq.(4), diverges from the generic GCN model. Typically, GCNs utilize the fundamental operation of neighborhood aggregation to consolidate information from neighboring nodes and generate the representation of the target node. In contrast, our backbone GNN model emphasizes edge-level representation learning by modeling the homophily distribution of each edge for aggregation to generate node representations. Therefore, relying on the intuition of distinctions in homophily distribution, as discussed in the Introduction (Figure 1(b)), our model places greater emphasis on edge-level homophily representation to address the classification of normal and anomalous nodes. Furthermore, Tables 1 and 2 in the Experiments consistently demonstrate that our backbone GNN model outperforms the GCN model, underscoring its effectiveness in handling anomaly detection on graphs.

---

> ### Author Response · Authors · 2023-11-18
>
> **2. As you said, GAD has low homophily distribution, whether the homophily-aware neighbourhood aggregation you propose will be effective for abnormal nodes, I deeply doubt it.**
>
>
> We sincerely apologize for any confusion that may have arisen, and we appreciate the opportunity to clarify this aspect of our work. In our study, we use the term "homophily" as a conceptual label, and it is important to note that "heterophily" could be interchangeably used in this context. Our model's primary goal with the homophily-aware neighborhood aggregation is to differentiate between the distinct distributions of homophily (or equivalently, heterophily) that are observed in the edges surrounding normal and anomalous nodes. Specifically, edges in the neighborhood of normal nodes typically exhibit high homophily (or low heterophily), whereas those around anomalous nodes tend to show low homophily (or high heterophily). The cross-entropy loss function in our model facilitates inverse predictions for normal and anomalous nodes. This means that the loss function strategically guides the edge representations surrounding a normal node to diverge from those around an anomalous node, aligning with their distinct homophily (or heterophily) distributions. Consequently, edges with high homophily around normal nodes share similar representations, while those with low homophily around anomalous nodes form analogous representations. This approach is critical for enhancing the discriminatory power of our model between normal and anomalous nodes, aligning with the core objectives of our research.
>
> **3. I don't find the definition of Vhq in Section 2, what is the meaning of Vun?**
>
> Thank you for highlighting the need for clarity regarding the definition of $V_{hq}$ and $V_{un}$ in our paper. We appreciate the opportunity to elucidate this aspect of our work. $V_{un}$ is defined as the set of unlabeled nodes.
> In the paragraph preceding Eq.(3), $V_{hq}$ represents the set of high-quality nodes, defined as $V_{hq}=\\{v \mid v\in V_{un} \land \exists \mathbf{p}_v[k]\ge\tau\\}$. This set comprises unlabeled nodes for which there exists at least one predicted score in their predictions greater than the threshold $\tau$. We have clarified this definition in our paper.

---

> ### Author Response · Authors · 2023-11-18
>
> **4. Please compare your learnable augmentation with other augmentation methods.**
>
> Thank you for your insightful suggestion.
> In Appendix E.4, we benchmark our learnable data augmentation against several stochastic augmentation methods. Specifically, we substitute our augmentation module with widely-recognized techniques in the consistency training framework, including *DropNode* (Feng et al., 2020a), *DropEdge* (Rong et al., 2020), and *Dropout* (Srivastava et al., 2014). We administer Dropout to both input features, denoted as *DropInput*, and intermediate features, referred to as *DropHidden*. It is noteworthy that DropHidden serves as a non-learnable analogue to our augmentation module. For a fair comparison, we determine the optimal drop rate for each method within the interval of $(0, 0.5]$, with a step size of $0.1$. We list the optimal drop rate in Table 6. For your ease of reference, we also reported these optimal rates in the table below.
>
> | Methods| Amazon | YelpChi | T-Finance | T-Social |
> | --- | --- | --- | --- | --- |
> | ConsisGAD (DropInput) | 0.1 | 0.1 | 0.1 | 0.4 |
> | ConsisGAD (DropNode) | 0.2 | 0.1 | 0.1 | 0.5 |
> | ConsisGAD (DropEdge) | 0.2 | 0.1 | 0.1 | 0.3 |
> | ConsisGAD (DropHidden) | 0.3 | 0.1 | 0.2 | 0.1 |
>
> The outcomes, depicted in Figure 9 through the three metrics, underscore the preeminence of our proposed learnable data augmentation over traditional stochastic alternatives.
> A salient observation is that among conventional augmentation methods, employing DropHidden in consistency training usually yields more substantial and stable performance improvements across all datasets. This empirical insight substantiates our decision to integrate learnable data augmentation at intermediate states and suggests the potential exploration of applying such augmentation to graph topology (Xia et al., 2022). For your easy reference, we also summarize the numerical results in the following table.
> | Dataset    | Metric   | ConsisGAD | ConsisGAD (DropInput) | ConsisGAD (DropNode) | ConsisGAD (DropEdge) | ConsisGAD (DropHidden) |
> | --- | --- | --- | --- | --- | --- | --- |
> | Amazon     | AUROC    | **93.91±0.58** | 89.16±0.53 | 91.47±0.73 | 92.63±0.45 | 92.71±0.38 |
> |            | AUPRC    | **83.33±0.34** | 74.01±1.30 | 77.84±1.03 | 80.22±0.89 | 81.03±1.00 |
> |            | Macro F1 | **90.03±0.53** | 86.22±0.93 | 85.58±0.68 | 88.03±0.81 | 87.78±0.91 |
> | YelpChi    | AUROC    | **83.36±0.53** | 81.68±0.21 | 79.85±0.26 | 80.70±0.47 | 80.27±0.39 |
> |            | AUPRC    | **47.33±0.58** | 42.14±0.52 | 40.14±0.67 | 43.54±0.27 | 43.50±0.21 |
> |            | Macro F1 | **69.72±0.30** | 68.22±0.37 | 66.31±0.47 | 67.34±0.29 | 67.32±0.44 |
> | T-Finance  | AUROC    | **95.33±0.30** | 95.17±0.23 | 95.07±0.40 | 95.13±0.49 | 94.58±0.45 |
> |            | AUPRC    | **86.63±0.44** | 84.87±0.61 | 84.78±0.31 | 85.43±0.47 | 85.59±0.43 |
> |            | Macro F1 | **90.97±0.63** | 90.16±0.33 | 90.02±0.66 | 90.16±1.02 | 90.31±1.05 |
> | T-Social   | AUROC    | **94.31±0.20** | 94.23±0.16 | 94.04±0.34 | 94.10±0.25 | 94.13±0.32 |
> |            | AUPRC    | **58.38±2.10** | 55.32±2.00 | 54.00±2.11 | 55.33±2.39 | 53.58±2.41 |
> |            | Macro F1 | **78.08±0.54** | 76.86±0.98 | 76.35±0.71 | 77.75±0.57 | 77.48±1.03 |

---

> ### Author Response · Authors · 2023-11-18
>
> **5. Please give complexity analysis.**
>
> As illustrated in Algorithm 1, our model is composed of two primary components: consistency training and the training of the learnable data augmentation module. These components operate iteratively in each iteration of the process. This approach potentially incurs a higher computational cost compared to the standard training procedures of graph neural networks. In this part, we give a detailed complexity analysis and analyze the possibility of using it on large graphs.
>
> At the outset, we focus on analyzing the complexity of our backbone GNN model, as outlined in Equation (4). This model is a critical component in Algorithm 1 for calculating node embeddings. Considering a target node $v$, in the first GNN layer, the function $\delta(\cdot,\cdot;\theta_{\delta})$ is implemented as $\text{MLP}(\mathbf{h}^{l-1}_v||\mathbf{h}^{l-1}_u)$. A typical example of this MLP is $\sigma(\mathbf{W}(\mathbf{h}^{l-1}_v \|\| \mathbf{h}^{l-1}_u)+\mathbf{b})$. The computational complexity in this first layer is $O(2dd_X+\bar{N}d+2d)$, where $d_X$ represents the dimension of the input feature vector, $d$ the intermediate dimension of the embeddings, and $\bar{N}$ the average node degree. Each subsequent layer contributes an additional complexity of $O(2d^2+\bar{N}d+2d)$. Given $L$ total GNN layers, the overall complexity of the GNN backbone sums up to $O(2dd_X+\bar{N}d+2d+(L-1)(2d^2+\bar{N}d+2d))=O(2dd_X+2Ld^2+2Ld+L\bar{N}d-2d^2)$.
> In every iteration of Algorithm 1, we sample a batch of labeled and unlabeled nodes for subsequent computations. These batches are of sizes $B$ (for labeled nodes) and $\mu B$ (for unlabeled nodes), respectively. We can split the whole process into three step as follows.
> - We begin by selecting high-quality nodes from the sampled batch of unlabeled nodes. In line 6, the GNN backbone introduces a complexity of $O(2dd_X+2Ld^2+2Ld+L\bar{N}d-2d^2)$, as previously analyzed. The prediction step in line 7 has a complexity of $O(Kd+2K)$, where $K$ is the number of classes. Lines 8-10 involve checking the high-quality nodes with a complexity of $K$. Overall, for a batch size of $B$, lines 5-10 collectively result in a complexity of $O(\mu B(2dd_X+2Ld^2+2Ld+L\bar{N}d-2d^2+Kd+3K))$.
> - Each high-quality node is augmented and predicted, forming the basis for the consistency and diversity loss calculations. For line 12, the complexity of the Sharpen function in Algorithm 2 involves $O(8\lfloor \xi d \rfloor d+d)$ across its steps. Consequently, line 12 in Algorithm 1 incurs a complexity of $O(8\lfloor \xi d \rfloor d+d^2+3d)$. Line 13, similar to line 7, involves a complexity of $O(Kd+2K)$. Line 14 includes forming both the consistency and diversity losses, with complexities of $O(\mu BK^2)$ and $O(2\mu Bd)$, respectively. Summing up, lines 11-14 entail a complexity of $O(\mu B(8\lfloor \xi d \rfloor d+d^2+3d + Kd+2K) + \mu BK^2 + 2\mu Bd)$.
> - The consistency training involves complexity calculations similar to the earlier sections. Lines 17 and 18, as previously analyzed, involve a complexity of $O(2dd_X+2Ld^2+2Ld+L\bar{N}d-2d^2+Kd+2K)$. Lines 19 and 20, pertaining to the loss calculations, have complexities of $O(BK^2)$ and $O(\mu BK^2)$, respectively. Thus, the total complexity for lines 16-20 over $B$ iterations is $O(B(2dd_X+2Ld^2+2Ld+L\bar{N}d-2d^2+Kd+2K)+BK^2+\mu BK^2)$.
>
> In each iteration of Algorithm 1, we aggregate the complexities from the previous sections to derive the overall complexity. This can be expressed as $O(2B(\mu+1)dd_X + B(2\mu L+2L-\mu-2)d^2 + (2\mu BL+\mu BL\bar{N}+\mu BK+8\mu B\lfloor \xi d \rfloor+5\mu B+K\mu B+2BL+BL\bar{N}+BK)d + 5\mu BK+2\mu BK^2+2BK+BK^2)$.
>
> Particularly in our anomaly detection scenario, where typically $K = 2$ due to the binary classification nature, and $L$ generally ranges from 1 to 3 as the number of GNN layers, the complexity of our model is predominantly influenced by the feature dimension $d_X$, the hidden dimension $d$, and the average node degree $\bar{N}$. This indicates that with appropriately set hyper-parameters, our model shows promising potential for application on large-scale graphs. To enhance the clarity of our paper, we have included this detailed complexity analysis in Appendix A of our revised manuscript.
>
>
> We hope that these clarifications and additional details address your concerns and strengthen the paper's contributions. Your insightful feedback has been instrumental in refining the quality and coherence of our manuscript. Should there be any more comments or issues from your side, we remain open and willing to make the requisite modifications.

---

> ### Author Response · Authors · 2023-11-21
> **A Kind Reminder to Reviewer tbTo**
>
> Dear Reviewer tbTo,
>
> Thank you once again for your insightful feedback on our submission. We would like to remind you that the discussion period is concluding. To facilitate your review, we have provided a concise summary below, outlining our responses to each of your concerns:
>
> - **Part 1**: We provided further details on how our homophily-aware neighborhood aggregation (Eq.4) achieves edge-level homophily representation and explained its distinctions from GCN aggregation.
> - **Part 2**: We offered additional clarifications on our homophily-aware neighborhood aggregation.
> - **Part 3**: We clarified the definitions of $V_{hq}$ and $V_{un}$.
> - **Part 4**: We conducted a comparative analysis of our learnable data augmentation against four stochastic augmentation methods for graph anomaly detection.
> - **Part 5**: We presented a detailed complexity analysis of our algorithm.
>
> We are grateful for your insightful comments and are eager to confirm whether our responses have adequately addressed your concerns. We look forward to any additional input you may provide.
>
> Warm regards, \
> The Authors of Submission 9315.

---

> ### Author Response · Authors · 2023-11-23
> **A Further Kind Reminder to Reviewer tbTo**
>
> Thanks a lot for your time in reviewing and insightful comments. We sincerely understand you are busy. But since the discussion due is approaching, would you mind checking the response to confirm where you have any further questions?
>
> We are looking forward to your reply and happy to answer your further questions.
>
> Warm regards, \
> The Authors of Submission 9315

---

### Official Review · Reviewer_RZbU · 2023-11-03

**Soundness:** 3 good
**Presentation:** 3 good
**Contribution:** 3 good
**Rating:** 8
**Confidence:** 4

**Summary:**

This paper delves into the challenge of graph anomaly detection with limited supervision. To address this, the authors introduce ConsisGAD, a novel model rooted in consistency training. ConsisGAD leverages variance in homophily distribution between normal and anomalous nodes to craft an effective GNN backbone. The authors also propose a learnable data augmentation mechanism for consistency training using unlabeled data. The paper presents comprehensive experiments on benchmark datasets to validate the effectiveness of the proposed model.

**Strengths:**

1. The paper is well-structured and presented in a clear manner. It effectively communicates the motivation behind the proposed model, making it accessible to readers.

2. The proposed model, comprising homophily-aware neighborhood aggregation and consistency training with learnable data augmentation, is innovative. The homophily-aware aggregation method is straightforward yet effective, especially for edge-level homophily representation, enhancing the anomaly detection process. The proposed label consistency and distribution diversity concepts are intriguing, guiding the optimization of learnable data augmentation.

3. The paper demonstrates a thorough evaluation through extensive experiments, which strengthens the empirical foundation and validates the performance across different datasets.

**Weaknesses:**

Weaknesses:

1. The paper should acknowledge related works that are pertinent to the proposed learnable data augmentation, such as [a] and [b]. It is crucial to cite and discuss the distinctions between these works and the proposed approach, providing readers with a clear understanding of the novel contributions made by this study.

2. The paper predominantly explores the concept of applying learnable data augmentation for graph anomaly detection. While this is valuable, investigating its applicability in broader graph learning tasks, such as node classification with contrastive learning, could significantly expand its scope. For example, how about its benefits to generic graph contrastive learning tasks, compared to existing contrastive techniques?

3. While consistency training might usually be deployed on unlabeled data, I wonder if it would be beneficial to utilize labeled data for consistency training as well. Specifically, labeled data has exact labels, which might provide effective information for consistency training the model in dealing with the taks of graph anomaly detection.


[a] Graph Contrastive Learning Automated. Yuning You, Tianlong Chen, Yang Shen, Zhangyang Wang. ICML 2021
[b] Graph Contrastive Learning with Adaptive Augmentation. Yanqiao Zhu, Yichen Xu, Feng Yu, Qiang Liu, Shu Wu, Liang Wang. WWW 2021.

**Questions:**

Please see the Weaknesses

---

> ### Author Response · Authors · 2023-11-18
>
> Dear Reviewer RZbU,
>
> Thank you for your insightful comments and constructive feedback on our submission 9315. We appreciate the time and effort you have invested in reviewing our work. Below, we address each of your comments to clarify and improve our manuscript:
>
> **1. The paper should acknowledge related works that are pertinent to the proposed learnable data augmentation, such as [a] and [b]. It is crucial to cite and discuss the distinctions between these works and the proposed approach, providing readers with a clear understanding of the novel contributions made by this study.**
>
> We acknowledge the importance of referencing and discussing relevant related works. In Appendix F of our revised manuscript, we have included a discussion comparing our work with the studies of [a] and [b]. For your convenience, we summarize the distinctions between these two works and our proposed learnable data augmentation:
>
> - **JOint Augmentation Optimization (JOAO) [a]**: This work learns a sampling distribution to sample augmentation pairs for graph contrastive learning. It instantiates the learning process as a min-max optimization process, iteratively optimizing the GNN encoder and the learnable sampling distribution. Our work is different from JOAO in two aspects:
>   - **Evaluation of augmentation quality**: JOAO is based on the adversarial training framework and assumes that a good augmentation will cause a high contrastive learning loss. Intuitively, increasing contrastive loss is to increase the distribution diversity of augmentations. However, the label consistency is overlooked, which may cause excessive noise injected into augmentations and lead to suboptimal performance, as indicated in prior studies [c,d]. In contrast, our work assumes that a good augmentation will increase the diversity of a node without changing its label (i.e., high label consistency). To this end, we formulate differentiable distribution diversity and label consistency metrics to evaluate augmentation quality and to guide the training of our augmentation module.
>   - **Learning objectives**: The learning objective of JOAO is to pick the optimal augmentation pair from a pre-defined pool of augmentation types for graph contrastive learning. It requires domain knowledge to manually construct and configure the augmentation pool for selection. By contrast, our work introduces a novel learnable augmentation module through learnable masking. Our module learns to synthesize node-specific augmentations, evolving dynamically with the dataset and GNN encoder.
>
> - **Graph Contrastive learning with Adaptive augmentation (GCA) [b]**: GCA introduces a general graph contrastive learning framework and generates augmented graphs by adaptively dropping edges and node features. The drop rates of edges and node features are based on node centrality scores, incorporating topological and semantic priors in a graph. The node centrality scores include degree centrality, eigenvector centrality, and PageRank centrality. Our work is different from GCA in two aspects:
>   - **Evaluation of augmentation quality**: GCA assumes that a good augmentation should drop unimportant edges and features while keeping important ones. The importance is measured through node centrality scores. In this way, edges whose ending nodes have low centrality scores are more likely to be dropped, and feature dimensions appearing less frequently on important nodes are more likely to be masked. However, this heuristic approach may not guarantee high consistency and diversity in augmentations. On the other hand, our proposed model does not rely on node importance metrics but focuses on maximizing distribution diversity and label consistency in augmentations.
>   - **Learning objectives**: The adaptive augmentation module in GCA, when presented with an input graph, remains fixed and non-learnable throughout the training process. This is due to the static nature of node centrality scores, resulting in unchanging dropping rates for edges and node features. Moreover, this module operates independently of the specific GNN encoder employed. On the contrary, our learnable augmentation module is dynamic, updating continuously throughout training to adapt to the dataset and the evolving GNN encoder.

---

> ### Author Response · Authors · 2023-11-18
>
> **2. The paper predominantly explores the concept of applying learnable data augmentation for graph anomaly detection. While this is valuable, investigating its applicability in broader graph learning tasks, such as node classification with contrastive learning, could significantly expand its scope. For example, how about its benefits to generic graph contrastive learning tasks, compared to existing contrastive techniques?**
>
> Thank you for your insightful suggestion regarding the potential applicability of our learnable data augmentation in generic graph contrastive learning tasks. We acknowledge the value of this idea. However, our current model is not straightforwardly applicable to such tasks due to the specific requirements of our label consistency metric. In graph contrastive learning, which focuses on pre-training a GNN encoder without specific label information, our method faces challenges, as it depends on label predictions for unlabeled nodes to measure label consistency.
>
> One potential approach to address this could involve clustering nodes in the input graph and then using these clusters as proxy labels to assess label consistency. Nevertheless, due to time constraints, we have not been able to explore this adaptation in the current study. We appreciate your suggestion and agree that expanding the applicability of our augmentation approach represents an exciting and valuable direction for future research. This would indeed enhance the versatility and impact of our learnable augmentation module and the proposed metrics for label consistency and distribution diversity. We are eager to explore this avenue in our future research, aiming to further broaden the scope of our work.

---

> ### Author Response · Authors · 2023-11-18
>
> **3. While consistency training might usually be deployed on unlabeled data, I wonder if it would be beneficial to utilize labeled data for consistency training as well. Specifically, labeled data has exact labels, which might provide effective information for consistency training the model in dealing with the task of graph anomaly detection.**
>
> Thank you for highlighting the potential benefits of utilizing labeled data for consistency training. In our current settings, both labeled and unlabeled data are used for consistency training, but labeled samples are treated as unlabeled, using their predicted labels instead of exact labels.
>
> Encouraged by your suggestion, we explore the impact of using exact labels for labeled nodes in consistency training. The results of this additional experiment are detailed in Appendix E.9 of our revised manuscript. For your convenience, we provide a summary in the table below:
>
> | Dataset    | Metric   | ConsisGAD (Predicted Label) | ConsisGAD (Exact Label) |
> |------------|----------|-----------------------------|-------------------------|
> | Amazon     | AUROC    | 93.91±0.58                  | 93.82±0.55              |
> |            | AUPRC    | 83.33±0.34                  | 83.16±0.46              |
> |            | Macro F1 | 90.03±0.53                  | 89.88±0.30              |
> | YelpChi    | AUROC    | 83.36±0.53                  | 82.91±0.36              |
> |            | AUPRC    | 47.33±0.58                  | 47.23±0.98              |
> |            | Macro F1 | 69.72±0.30                  | 69.44±0.29              |
> | T-Finance  | AUROC    | 95.33±0.30                  | 95.09±0.27              |
> |            | AUPRC    | 86.63±0.44                  | 86.94±0.13              |
> |            | Macro F1 | 90.97±0.63                  | 91.32±0.18              |
> | T-Social   | AUROC    | 94.31±0.20                  | 93.87±0.30              |
> |            | AUPRC    | 58.38±2.10                  | 56.54±1.86              |
> |            | Macro F1 | 78.08±0.54                  | 77.42±0.67              |
>
> Our findings indicate that using exact labels in consistency training does not significantly improve performance on most datasets. This lack of improvement may stem from the fact that labeled data has already been utilized in the cross-entropy loss, and their reuse in the consistency training does not contribute significantly to performance enhancement. Notably, a decrease in performance is observed in the T-Social dataset. We hypothesize this could be attributed to the model overfitting to label noise, which could be mitigated by the use of predicted labels.
>
> We hope these revisions to our submission adequately address your concerns and enhance its overall contributions. Your thoughtful feedback has been crucial in elevating the quality and coherence of our manuscript. Should you have additional comments or concerns, we remain receptive and prepared to undertake any further modifications.
>
> [a] Graph Contrastive Learning Automated. Yuning You, Tianlong Chen, Yang Shen, Zhangyang Wang. ICML 2021.
>
> [b] Graph Contrastive Learning with Adaptive Augmentation. Yanqiao Zhu, Yichen Xu, Feng Yu, Qiang Liu, Shu Wu, Liang Wang. WWW 2021.
>
> [c] Instance adaptive adversarial training: Improved accuracy tradeoffs in neural nets. Yogesh Balaji, Tom Goldstein, and Judy Hoffman. arXiv preprint arXiv:1910.08051, 2019.
>
> [d] Robustness May Be at Odds with Accuracy. Dimitris Tsipras, Shibani Santurkar, Logan Engstrom, Alexander Turner, and Aleksander Madry. ICLR, 2018.

---

> ### Author Response · Authors · 2023-11-21
> **A Kind Reminder to Reviewer RZbU**
>
> Dear Reviewer RZbU,
>
> Thank you once again for your insightful feedback on our submission. We would like to remind you that the discussion period is concluding. To facilitate your review, we have provided a concise summary below, outlining our responses to each of your concerns:
>
> - **Part 1**: We elucidated the distinctions between our learnable data augmentation and other automated data augmentation methods, such as JOAO and GCA.
> - **Part 2**: We explored the application of our learnable augmentation module in generic graph contrastive learning tasks.
> - **Part 3**: We conducted further analysis on the effectiveness of utilizing the exact labels of labeled nodes in consistency training.
>
> We are grateful for your insightful comments and are eager to confirm whether our responses have adequately addressed your concerns. We look forward to any additional input you may provide.
>
> Warm regards, \
> The Authors of Submission 9315.

---

> ### Author Response · Authors · 2023-11-23
> **A Further Kind Reminder to Reviewer RZbU**
>
> Thanks a lot for your time in reviewing and insightful comments. We sincerely understand you are busy. But since the discussion due is approaching, would you mind checking the response to confirm where you have any further questions?
>
> We are looking forward to your reply and happy to answer your further questions.
>
> Warm regards, \
> The Authors of Submission 9315

---

### Official Review · Reviewer_ZNLh · 2023-11-05

**Soundness:** 3 good
**Presentation:** 3 good
**Contribution:** 3 good
**Rating:** 8
**Confidence:** 4

**Summary:**

In this paper, the authors tackle the problem of anomaly detection in graphs by introducing a novel approach---consistency training coupled with a learnable data augmentation mechanism. Their method involves leveraging a simple GNN backbone that emphasizes edge-level homophily representation for anomaly detection. By exploiting disparities in homophily distribution, the proposed model achieves enhanced anomaly detection capabilities. Additionally, the authors introduce two evaluation metrics designed to optimize their innovative learnable data augmentation technique, guiding its effectiveness in data synthesis. In their experimental evaluation, the authors assess the performance of their model across four benchmark datasets commonly used for graph-based anomaly detection, along with a private dataset. The proposed model generally outperforms existing state-of-the-art baselines across three key evaluation metrics.

**Strengths:**

1. I find the proposed learnable data augmentation mechanism intriguing. Intuitively, this approach appears effective, as it can be finely tuned through the optimization of two core metrics: label consistency and data distribution. Through iterative optimization of these components---the GNN module and the learnable data augmentation module---it seems both elements can be well-trained, enhancing the overall performance of the model.

2. The node clusters displayed in Figure 3 provide a compelling visual representation of the effectiveness of the proposed GNN backbone. The well-separated node clusters, especially concerning different edge relations, serve as a strong validation of the necessity of leveraging distinctions in homophily distribution for anomaly detection. This visualization underscores the robustness of the proposed method.

3. The paper is skillfully composed, ensuring clarity and coherence in conveying the research concepts. Additionally, the experiments conducted are generally comprehensive, offering substantial evidence to support the effectiveness of the proposed model.

**Weaknesses:**

1. Regarding consistency training, the proposed model employs thresholds to identify "high-quality nodes" from the unlabeled data for training. This step, akin to semi-supervised learning, assigns synthetic labels to unlabeled data for consistency training. The performance of these "high-quality nodes" is pivotal for downstream iterative training, emphasizing the significance of their accuracy. Hence, the authors should analyze the actual quality of these "high-quality nodes" and the accuracy of their predicted labels. This aspect holds significant importance in evaluating the reliability and effectiveness of the model.

2. The paper introduces essential hyperparameters, including $\alpha$ and $\xi$. While the authors provide their specific settings in the experiments, the impact of these parameters on the performance of the model remains unexplored. The authors should perform a sensitivity analysis for the crucial hyperparameters in the proposed model.

3. The proposed GNN backbone shows promise for anomaly detection by capturing distinctions in the homophily distribution of nodes in graphs. However, it raises intriguing questions about its performance on graphs with varying levels of heterophily. Specifically, how well does the proposed GNN backbone perform on graphs with low or high heterophily when applied to generic multi-class node classification tasks?

**Questions:**

1. How about the correctness of the "high-quality nodes"?

2. The authors should perform a sensitivity analysis for the crucial hyperparameters in the proposed model.

3. Can the proposed GNN backbone effectively handle generic multi-class node classification tasks?

---

> ### Author Response · Authors · 2023-11-18
>
> Thank you for your thorough review and insightful comments on our submission. We appreciate the time you invested in evaluating our work and offering constructive feedback. Below, we address your comments to clarify and improve our manuscript.
>
> **1. How about the correctness of the "high-quality nodes"?**
>
> Thanks for highlighting the significance of evaluating the quality of "high-quality nodes" in our consistency training approach. Considering your valuable suggestion, we have conducted a detailed experiment to assess this aspect and include the results in Appendix E.6 of our revised manuscript. In this experiment, we visualize Macro F1 scores of "high-quality nodes" at each epoch during training and compare these with scores from the test set. For your reference, we report the scores with an interval of 10 epochs in the following table. For more details, please kindly refer to the Appendix E.6.
>
> |                    | Epoch 10 | Epoch 20 | Epoch 30 | Epoch 40 | Epoch 50 | Epoch 60 | Epoch 70 | Epoch 80 | Epoch 90 | Epoch 100 |
> | :----------------: | :------: | :------: | :------: | :------: | :------: | :------: | :------: | :------: | :------: | :-------: |
> |       Amazon       |          |          |          |          |          |          |          |          |          |           |
> | High-quality Nodes |  90.79   |  88.89   |  91.06   |  92.18   |  90.91   |  90.76   |  90.76   |  91.43   |  89.99   |   89.29   |
> |     Test Nodes     |  87.56   |  88.13   |  90.10   |  90.96   |  90.98   |  90.92   |  90.04   |  90.60   |  90.07   |   90.43   |
> |       Diff.        |   3.23   |   0.76   |   0.96   |   1.23   |  -0.07   |  -0.17   |   0.72   |   0.83   |  -0.08   |   -1.14   |
> |      YelpChi       |          |          |          |          |          |          |          |          |          |           |
> | High-quality Nodes |  67.23   |  69.23   |  68.09   |  69.11   |  68.49   |  68.98   |  69.21   |  70.02   |  69.28   |   68.76   |
> |     Test Nodes     |  66.72   |  67.56   |  68.49   |  67.76   |  68.13   |  68.12   |  67.42   |  67.96   |  68.25   |   68.42   |
> |       Diff.        |   0.51   |   1.68   |  -0.40   |   1.35   |   0.36   |   0.86   |   1.79   |   2.06   |   1.03   |   0.33    |
> |     T-Finance      |          |          |          |          |          |          |          |          |          |           |
> | High-quality Nodes |  90.82   |  90.27   |  90.15   |  90.12   |  88.50   |  91.33   |  91.11   |  91.48   |  91.14   |   91.09   |
> |     Test Nodes     |  89.28   |  90.22   |  89.39   |  90.44   |  91.16   |  91.05   |  90.90   |  90.87   |  90.62   |   90.39   |
> |       Diff.        |   1.54   |   0.05   |   0.76   |  -0.31   |  -2.66   |   0.28   |   0.21   |   0.62   |   0.52   |   0.71    |
> |      T-Social      |          |          |          |          |          |          |          |          |          |           |
> | High-quality Nodes |  75.68   |  76.35   |  76.92   |  75.82   |  76.09   |  77.53   |  75.48   |  74.07   |  76.11   |   73.59   |
> |     Test Nodes     |  72.14   |  73.56   |  76.07   |  74.17   |  74.23   |  76.10   |  73.13   |  71.52   |  73.81   |   72.58   |
> |       Diff.        |   3.53   |   2.80   |   0.85   |   1.65   |   1.87   |   1.43   |   2.35   |   2.54   |   2.30   |   1.01    |
>
> The result demonstrates that "high-quality nodes" consistently achieve higher Macro F1 scores than test nodes across various datasets. On Amazon, YelpChi, T-Finance, and T-Social, the average performance improvement of "high-quality nodes" over test nodes is 1.01%, 0.83%, 0.71%, and 0.64%, respectively. Generally, as depicted in Figure 10, both "high-quality nodes" and test nodes exhibit a gradual increase in performance, with the former driving the improvement of the latter. An exception lies in the T-Social dataset, on which we observe some performance fluctuations. This is possibly due to the much smaller batch size and training data relative to the whole dataset size. Overall, our analysis showcases the accuracy of the predicted labels and the robustness of our selection criteria for "high-quality nodes".

---

> ### Author Response · Authors · 2023-11-18
>
> **2. The authors should perform a sensitivity analysis for the crucial hyperparameters in the proposed model.**
>
> Taking your valuable advice, we have performed a sensitivity analysis on hyper-parameters and included the results in Appendix E.7 of our revised manuscript. We vary the weights of the label consistency loss $\alpha$ among \{0.1, 0.2, 0.5, 1.0, 2.0, 5.0, 10.0\} and the drop ratio $\xi$ among \{0.0, 0.1, 0.2, 0.3, 0.4, 0.5\}, and measure the corresponding model performance across the Amazon, YelpChi, and T-Finance datasets. For your ease of reference, we have summarized the results in the tables below. Additionally, we have included in Appendix E.7 a sensitivity analysis of the normal threshold $\tau_n$ and anomalous threshold $\tau_a$, providing a more comprehensive understanding of our model's behavior under different hyper-parameter settings.
>
> | $\alpha$  |    0.1     |    0.2     |      0.5       |      1.0       |    2.0     |      5.0       |    10.0    |
> | :-------: | :--------: | :--------: | :------------: | :------------: | :--------: | :------------: | :--------: |
> |  Amazon   |            |            |                |                |            |                |            |
> |   AUROC   | 93.16±0.56 | 93.55±0.56 |   93.68±0.63   | **93.91±0.58** | 93.73±0.54 |   92.89±0.60   | 93.31±0.58 |
> |   AUPRC   | 82.47±0.48 | 82.38±0.35 |   82.76±0.51   | **83.33±0.34** | 83.13±0.69 |   80.37±0.84   | 79.21±1.04 |
> | Macro F1  | 89.50±0.81 | 89.27±0.66 |   89.03±0.69   | **90.03±0.53** | 89.77±0.27 |   89.56±0.40   | 88.65±0.65 |
> |  YelpChi  |            |            |                |                |            |                |            |
> |   AUROC   | 81.86±0.25 | 81.76±0.41 | **83.36±0.53** |   82.16±0.63   | 82.94±0.62 |   82.01±0.81   | 81.34±0.50 |
> |   AUPRC   | 45.33±0.33 | 45.00±0.52 | **47.33±0.58** |   45.49±1.08   | 46.27±0.82 |   44.88±1.19   | 43.47±0.26 |
> | Macro F1  | 68.73±0.33 | 68.63±0.52 | **69.72±0.30** |   68.81±0.37   | 69.11±0.50 |   68.55±0.43   | 68.25±0.36 |
> | T-Finance |            |            |                |                |            |                |            |
> |   AUROC   | 95.06±0.31 | 95.10±0.22 |   95.15±0.48   |   95.01±0.40   | 95.26±0.17 | **95.33±0.30** | 95.13±0.45 |
> |   AUPRC   | 84.77±0.47 | 85.37±0.37 |   85.71±0.77   |   85.88±1.23   | 86.10±0.75 | **86.63±0.44** | 83.87±0.39 |
> | Macro F1  | 89.57±0.68 | 89.74±0.41 |   90.29±0.69   |   90.93±0.55   | 90.74±0.68 | **90.97±0.63** | 89.17±0.84 |
>
>
> |   $\xi$   |    0.0     |      0.1       |      0.2       |      0.3       |    0.4     |    0.5     |
> | :-------: | :--------: | :------------: | :------------: | :------------: | :--------: | :--------: |
> |  Amazon   |            |                |                |                |            |            |
> |   AUROC   | 92.07±0.29 |   92.93±0.15   |   93.54±0.47   | **93.91±0.58** | 93.05±0.82 | 92.43±1.13 |
> |   AUPRC   | 78.79±0.42 |   80.20±0.74   |   81.58±0.31   | **83.33±0.34** | 81.66±1.33 | 80.46±1.82 |
> | Macro F1  | 87.25±1.21 |   87.26±0.99   |   88.93±1.15   | **90.03±0.53** | 89.38±0.90 | 89.48±0.87 |
> |  YelpChi  |            |                |                |                |            |            |
> |   AUROC   | 80.36±0.62 | **83.36±0.53** |   82.45±0.65   |   81.93±0.83   | 81.66±0.63 | 81.82±1.02 |
> |   AUPRC   | 42.60±1.18 | **47.33±0.58** |   45.47±1.12   |   44.59±0.99   | 43.80±0.65 | 43.65±1.28 |
> | Macro F1  | 67.22±0.76 | **69.72±0.30** |   68.89±0.66   |   68.21±0.72   | 67.87±0.27 | 68.10±0.58 |
> | T-Finance |            |                |                |                |            |            |
> |   AUROC   | 95.18±0.56 |   95.24±0.27   | **95.33±0.30** |   95.14±0.33   | 95.13±0.39 | 95.20±0.24 |
> |   AUPRC   | 83.57±0.64 |   86.55±0.82   | **86.63±0.44** |   86.39±0.88   | 85.29±0.71 | 85.10±0.73 |
> | Macro F1  | 88.79±1.21 |   90.89±0.73   | **90.97±0.63** |   90.73±1.03   | 89.69±1.14 | 88.98±1.09 |
>
> For the weight of the label consistency loss $\alpha$, our results indicate that an $\alpha$ value of around 1.0 yields optimal performance for the Amazon and YelpChi datasets, whereas a value of around 5.0 is preferable for the T-Finance dataset. Overall, the model performance remains stable with a moderate $\alpha$ value. If $\alpha$ is set to an excessively high value, the performance will be greatly impaired. This is likely due to the diminished diversity in generated augmentations, which is crucial for effective consistency training.
>
> Regarding the drop ratio $\xi$, our results reveal that maintaining $\xi$ within a range of 0.1 to 0.3 offers consistent benefits across all datasets. A smaller $\xi$ reduces ConsisGAD to its backbone model, limiting its ability to harness unlabeled data effectively. Conversely, a higher $\xi$ could lead to significant information loss, making it hard for the model to discern valuable patterns.

---

> ### Author Response · Authors · 2023-11-18
>
> **3. Can the proposed GNN backbone effectively handle generic multi-class node classification tasks?**
> Pursuant to your advice, we have extended our experiments to include graphs with low and high heterophily and discussed the results in Appendix E.8 of our revised manuscript. In this experiment, we adhere to the experimental procedure of a recent work, the Feature Selection Graph Neural Network (FSGNN) [a], and use their public repository for experimentation. Our evaluation encompasses three homophily graphs and six heterophily graphs, with homophily ratios ranging from 0.11 to 0.81. This range provides a comprehensive spectrum for performance assessment of our model. For your easy reference, we list the dataset statistics in the following table:
>
> |                |      Cora      |  Citeseer  |     Pubmed     |   Chameleon    |   Wisconsin    |     Texas      |    Cornell     |    Squirrel    |     Actor      |
> | :------------- | :------------: | :--------: | :------------: | :------------: | :------------: | :------------: | :------------: | :------------: | :------------: |
> | Hom. ratio     |      0.81      |    0.74    |      0.80      |      0.23      |      0.21      |      0.11      |      0.30      |      0.22      |      0.22      |
> | \# Nodes       |     2,708      |   3,327    |     19,717     |     2,277      |      251       |      183       |      183       |     5,201      |     7,600      |
> | \# Edges       |     5,278      |   4,732    |     44,338     |     36,101     |      499       |      309       |      295       |    198,353     |     26,659     |
> | \# Features    |     1,433      |   3,703    |      500       |     2,325      |     1,703      |     1,703      |     1,703      |     2,089      |      932       |
> | \# Classes     |       7        |     6      |       3        |       4        |       5        |       5        |       5        |       5        |       5        |
>
> We maintain a consistent architecture for our GNN backbone, setting it to two layers, and conduct fine-tuning for learning rate and weight decay on each dataset. The model is evaluated in ten publicly available data splits, and the mean test accuracy is reported. For your convenience, we list the results as below:
>
> |                |      Cora      |  Citeseer  |     Pubmed     |   Chameleon    |   Wisconsin    |     Texas      |    Cornell     |    Squirrel    |     Actor      | Mean Acc. |
> | :------------- | :------------: | :--------: | :------------: | :------------: | :------------: | :------------: | :------------: | :------------: | :------------: | :-------: |
> | GCN            |   87.28±1.26   | 76.68±1.64 |   87.38±0.66   |   59.82±2.58   |   59.80±6.99   |   59.46±5.25   |   57.03±4.67   |   36.89±1.34   |   30.26±0.79   |   61.62   |
> | GAT            |   82.68±1.80   | 75.46±1.72 |   84.68±0.44   |   54.69±1.95   |   55.29±8.71   |   58.38±4.45   |   58.92±3.32   |   30.62±2.11   |   26.28±1.73   |   58.55   |
> | GraphSAGE      |   86.90±1.04   | 76.04±1.30 |   88.45±0.50   |   58.73±1.68   |   81.18±5.56   |   82.43±6.14   |   75.95±5.01   |   41.61±0.74   |   34.23±0.99   |   69.50   |
> | Cheby+JK       |   85.49±1.27   | 74.98±1.18 |   89.07±0.30   |   63.79±2.27   |   82.55±4.57   |   78.38±6.37   |   74.59±7.87   |   45.03±1.73   |   35.14±1.37   |   69.89   |
> | MixHop         |   87.61±0.85   | 76.26±1.33 |   85.31±0.61   |   60.50±2.53   |   75.88±4.90   |   77.84±7.73   |   73.51±6.34   |   43.80±1.48   |   32.22±2.34   |   68.10   |
> | GEOM-GCN       |     85.27      | **77.99**  |     90.05      |     60.90      |     64.12      |     67.57      |     60.81      |     38.14      |     31.63      |   64.05   |
> | GCNII          |   88.01±1.33   | 77.13±1.38 | **90.30±0.37** |   62.48±2.74   |   81.57±4.98   |   77.84±5.64   |   76.49±4.37   |      N/A       |      N/A       |     -     |
> | H2GCN-1        |   86.92±1.37   | 77.07±1.64 |   89.40±0.34   |   57.11±1.58   |   86.67±4.69   |   84.86±6.77   |   82.16±4.80   |   36.42±1.89   |   35.86±1.03   |   70.71   |
> | WRGAT          |   88.20±2.26   | 76.81±1.89 |   88.52±0.92   |   65.24±0.87   |   86.98±3.78   |   83.62±5.50   |   81.62±3.90   |   48.85±0.78   |   36.53±0.77   |   72.93   |
> | GPRGNN         | **88.49±0.95** | 77.08±1.63 |   88.99±0.40   |   66.47±2.47   |   85.88±3.70   |   86.49±4.83   |   81.89±6.17   |   49.03±1.28   |   36.04±0.96   |   73.37   |
> | FSGNN          |   88.23±1.17   | 77.40±1.93 |   89.78±0.38   | **78.95±0.86** | **88.43±3.22** | **87.57±4.86** | **87.84±6.19** | **74.10±1.89** |   35.75±0.96   | **78.67** |
> | CONSISGAD(GNN) |   86.32±1.72   | 75.83±1.81 |   89.39±0.34   |   44.82±2.96   |   86.47±4.51   |   83.24±5.77   |   83.51±6.89   |   33.08±0.79   | **37.38±1.55** |   68.89   |

---

> ### Author Response · Authors · 2023-11-18
>
> It is important to note that our proposed GNN backbone, ConsisGAD(GNN), is specifically tailored to exploit the homophily distribution differences between normal and anomalous nodes. Such a phenomenon is characteristic in the graph anomaly detection task (i.e., the imbalanced binary classification task) where normal nodes have high homophily distribution while anomalous nodes have low homophily distribution. This prominent distribution discrepancy lays down the foundation of our GNN backbone. When it comes to multi-class node classification tasks, the homophily-aware neighborhood aggregation (Equation (4)) in our GNN backbone will model the neighbor label distribution within a node's neighborhood. If this neighborhood label distribution exhibits distinguishable patterns for different classes, we would expect our backbone model to function well.
>
> From the results, we observe that our GNN backbone model performs comparably to existing baselines in the three homophily graphs. Our GNN backbone is built upon the Message-Passing Neural Network (MPNN) framework, which allows handling homophily information naturally. For heterophily graphs, our backbone model yields promising results in the Wisconsin, Texas, and Cornell datasets, but encounters challenges in the Chameleon, Squirrel, and Actor datasets. Our further investigations in Figure 15 reveal that, in Wisconsin, Texas, and Cornell, nodes of different classes have distinct neighbor label distribution within their neighborhood, which allows our backbone model to distinguish different classes effectively. However, such distinct patterns are absent in Chameleon, Squirrel, and Actor, explaining the struggle of our backbone model on them. Notably, despite a low accuracy on the Actor dataset, our model achieves a new state-of-the-art result, underscoring its potential in handling heterophily graphs in generic multi-class node classification tasks. We recognize the opportunity for further enhancements, particularly on datasets where distinct neighborhood patterns are less pronounced. Future research may explore synergies between our GNN backbone model and other advanced techniques to elevate performance on heterophily graphs.
>
> We hope these revisions and additional analyses address your concerns and strengthen the contribution of our work. We appreciate the constructive guidance you have provided, which has substantially improved the manuscript. If you have additional comments or concerns, we welcome your input and are ready to make any necessary adjustments.
>
> [a] Simplifying approach to node classification in graph neural networks. Sunil Kumar Maurya, Xin Liu, and Tsuyoshi Murata. Journal of Computational Science, 2022.

---

> ### Author Response · Authors · 2023-11-21
> **A Kind Reminder to Reviewer ZNLh**
>
> Dear Reviewer ZNLh,
>
> Thank you once again for your insightful feedback on our submission. We would like to remind you that the discussion period is concluding. To facilitate your review, we have provided a concise summary below, outlining our responses to each of your concerns:
>
> - **Part 1**: We conducted additional experiments to assess the quality of high-quality nodes. The results consistently demonstrated superior performance of high-quality nodes compared to test nodes.
> - **Part 2**: We conducted additional experiments to assess the sensitivity of our hyperparameters: the weight of the label consistency loss ($\alpha$) and the drop ratio ($\xi$).
> - **Part 3**: We evaluated our GNN backbone on a generic multi-class node classification task. The experiments comprised three homophily graphs and six heterophily graphs, with homophily ratios ranging from 0.11 to 0.81. Additionally, we conducted a detailed performance comparison analysis.
>
> We are grateful for your insightful comments and are eager to confirm whether our responses have adequately addressed your concerns. We look forward to any additional input you may provide.
>
> Warm regards, \
> The Authors of Submission 9315.

---

> ### Comment · Reviewer_ZNLh · 2023-11-21
> **increase score**
>
> Thanks for the authors' detailed rebuttal. It generally addresses my comments in the review. Thus, I increase my score and support this paper.

---

> ### Author Response · Authors · 2023-11-22
> **Appreciate Your Feedback**
>
> Dear Reviewer ZNLh,
>
> We sincerely value your constructive feedback, and we are delighted that our responses have effectively addressed your concerns. Your recognition of our efforts in the paper is greatly appreciated!
>
> Warm regards, \
> The Authors of Submission 9315.

---

### Author Response · Authors · 2023-11-23
**A Summary of Our Contributions and Responses**

Dear Reviewers and ACs,

As the discussion is ending in the next few hours, we are grateful for the insightful comments and valuable suggestions from all reviewers. We would really appreciate it if reviewer _RZbU_ and reviewer _tbTo_ could kindly re-evaluate their reviews considering our follow-up responses and manuscript revisions. In the following, we would like to summarize the contributions and responses of this paper again.

**Contributions**:

- To deal with graph anomaly detection with limited supervision, we propose a learnable augmentation module and optimize it via two metrics: label consistency and distribution diversity. This approach appears ```effective```, ```enhancing the overall performance of the model``` (_R ZNLh_), and ```innovative``` (_R RZbU_). The two proposed metrics are ```intriguing``` (_R RZbU_).
- We introduce a GNN backbone model that exploits the homophily distribution differences between normal and anomalous nodes for detection. This approach is ```straightforward yet effective``` (_R RZbU_). ```Visualization underscores the robustness of the proposed method``` (_R ZNLh_).
- Our paper is ```well-written``` (_R ZNLh_, _R RZbU_, _R tbTo_). Our experiments are generally ```comprehensive``` (_R ZNLh_), ```extensive``` (_R RZbU_), and ```sufficient``` (_R tbTo_).

**Responses and Revisions:**

- **For Reviewer _ZNLh_'s concerns**:
  - We conducted additional experiments to assess and analyze the quality of high-quality nodes.
  - We conducted additional experiments to analyze the sensitivity of our model to several important hyper-parameters.
  - We conducted additional experiments to evaluate and analyze the applicability of our GNN backbone model to generic multi-class node classification tasks.
- **For Reviewer _RZbU_' concerns**:
  - We made a detailed comparison between our learnable data augmentation module and existing automatic data augmentation techniques.
  - We explored and analyzed the extensions of our learnable augmentation module to generic graph contrastive learning tasks.
  - We conducted additional experiments to analyze the effectiveness of utilizing the exact labels of labeled nodes in consistency training.
- **For Reviewer _tbTo_'s concerns**:
  - We made clarifications on the architecture and motivations of our proposed model (Responses to Questions 1, 2, and 3).
  - We highlighted our comparative analysis of our learnable data augmentation against four stochastic augmentation methods in the paper.
  - We gave a comprehensive complexity analysis of our algorithm.

All modifications have been highlighted in blue in our revised manuscript. Thanks again for your efforts in reviewing our work, and we hope our responses adequately address any concerns about this work.

Warm regards, \
The Authors of Submission 9315

---

### Public Comment · ~Junwei_He1 · 2023-12-12
**Code and Dataset Sharing**

I've been looking at the results of your recent study and I'm quite impressed with what you've accomplished. Your experiments seem to show some really interesting findings that could be important for our field.

I do have a small request that I believe could help others appreciate your work as much as I do. Would it be possible for you to share the code and datasets you used? Making these available would make it easier for others (myself included) to understand and potentially replicate your work, which would really strengthen the impact of your findings.

I know sharing these resources can take extra effort, but I think it would be incredibly valuable to our community. It would help everyone get a better grasp of your methods and possibly inspire more research in this area.

Thanks a lot for considering this.

---

### Meta-Review · Area_Chair_PR2H · 2023-12-04

**Metareview:**

The majority of the reviewers were in strong support of acceptance (two scores of 8), and found novelty in the proposed learnable data augmentation approach.

**Justification For Why Not Higher Score:**

One reviewer gave a 5, but then did not engage in discussion.

**Justification For Why Not Lower Score:**

There is significantly strong support, with high confidence, from the majority of the reviewers.

---

### Decision · Program_Chairs · 2024-01-16

Accept (spotlight)